# COunterfactual Reasoning for Temporal EXplanations: Plausible and Robust Explanations for EEG-Based Seizure Detection

**Martina Zannotti**[*]                                                            *martina.zannotti@unicam.it*
*Computer Science Division, University of Camerino, Italy*
*Syeew s.r.l., Italy*

**Bardh Prenkaj**                                                                  *bardh.prenkaj@tum.de*
*Technical University of Munich, Germany*

**Marco Piangerelli**                                                              *marco.piangerelli@unicam.it*
*Computer Science Division, University of Camerino, Italy*
*Vici & C. S.p.A., Italy*

**Flavio Corradini**                                                               *flavio.corradini@unicam.it*
*Computer Science Division, University of Camerino, Italy*

**Gjergji Kasneci**                                                                *gjergji.kasneci@tum.de*
*Technical University of Munich, Germany*

**Reviewed on OpenReview:** *https://openreview.net/forum?id=FkHVmYnNS9*

## Abstract

Identifying the drivers of change in time-sensitive domains like healthcare is critical for reliable decision-making, yet explanations must account for both temporal dynamics and structural complexity. While counterfactual explanations are well-studied for static data, existing methods often fail in dynamic, spatio-temporal settings, producing implausible or temporally inconsistent explanations. To address this, we introduce COunterfactual Reasoning for Temporal EXplanations (CORTEX), a search-based explainer for multivariate time series modeled as spatio-temporal graphs, tailored to binary seizure detection from EEG recordings. CORTEX generates temporally robust and plausible counterfactuals by retrieving relevant past instances and sieving them via structural dissimilarity, temporal distance, and robustness. As a result of its design choices, when evaluated on clinical seizure detection data, CORTEX outperforms state-of-the-art methods by $3.73\times$ in validity and $6.32\times$ in fidelity, and achieves zero implausibility, demonstrating consistency and practical relevance. By shifting the focus from mere validity to plausible, time-consistent explanations, CORTEX enables more reliable, controllable counterfactual explanations.

## 1 Introduction

Understanding the temporal evolution of a model's prediction is critical for trust and accountability, especially in sensitive domains like healthcare, where abrupt changes in a patient's physiological state – e.g, those measured by Electroencephalography (EEG) – require timely intervention. Counterfactual explanations (Wachter et al., 2017) are a promising approach to guide potential interventions for the end-user, as they identify minimal and plausible changes to the input features that would lead the model to a different outcome. However, while common for static data, their application to multivariate time series is limited.

---

[*]Work done while at the Technical University of Munich.

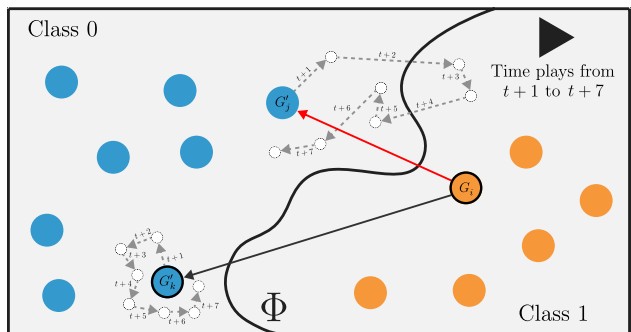

Figure 1: **Gist of CORTEX: When choosing counterfactuals in dynamic contexts, consider temporal evolution alongside similarity.** Given a graph $G_i$, $G'_j$ and $G'_k$ are two plausible counterfactual candidates. However, $G'_j$, as time goes by, has a "chaotic" trajectory; meanwhile, $G'_k$ has a more confined movement. We prefer $G'_k$ as a counterfactual for $G_i$ since it is likely to remain in the counterfactual class over time (i.e., to be more stable, temporally robust). To simplify the presentation, we have only depicted the change trajectories of $G'_j$ and $G'_k$ over time.

Existing methods fall short by focusing on feature modifications without preserving underlying implicit correlations (Ates et al., 2021) or by restricting modifications to predefined deviation intervals (Yamaguchi et al., 2024). These approaches neither reflect the spatio-temporal structure nor focus on the structural differences that drive changes in model outcomes, making them fundamentally inadequate for dynamic, autoregressive contexts where modifications to past observations propagate and influence future trajectories.

To effectively illustrate the distinction between static and spatio-temporal counterfactual explanations in our scenario, we provide Example 1 (best viewed in color):

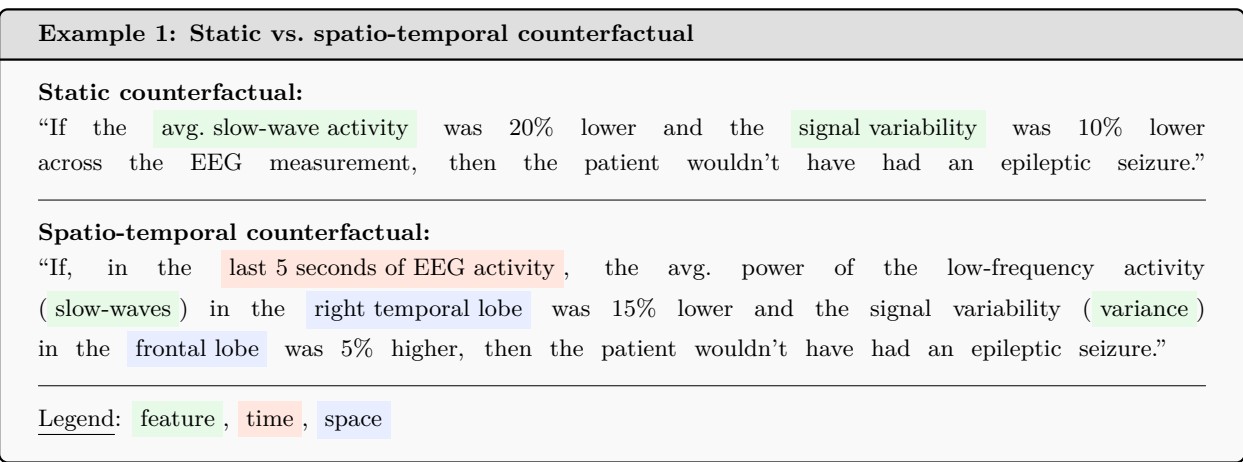

This highlights the complexity of the spatio-temporal counterfactual: i.e., it must not only address three dimensions of explainability (feature, time, and space) but also explicitly process data inherently structured by these dimensions and account for temporal dependencies from past to future.

In this paper, we address the challenge of producing robust temporal graph counterfactual explanations for multivariate time series, tailored to binary epileptic seizure detection from EEG data. We aim to identify the changes required to transition from an anomalous to a normal state, terminating the epileptic episode and restoring normal brain activity, which is a clinically relevant objective.[1] For this purpose, we propose a novel search-based explainer, COunterfactual Reasoning for Temporal EXplanations (CORTEX). To account for the relationships among features, time, and space, we represent time series as spatio-temporal graphs and

---

[1] These counterfactuals are intended to highlight structural differences and patterns in the data, and should not be interpreted as clinical recommendations.

generate counterfactuals over them. Although our approach is not restricted to graph-based representations, modeling time series as spatio-temporal graphs allows us to explicitly encode interactions among features over time, which are difficult to capture with purely tabular- or sequence-based representations (Corradini et al., 2026).

Since the underlying dynamics of EEG readings are unknown, it may be unfeasible to generate plausible counterfactuals from scratch. Hence, CORTEX draws counterfactuals from previously observed robust states (see Figure 1), and then ranks them according to structural similarity, temporal distance, and temporal robustness.

This paper makes the following key contributions:

**(1) A new paradigm for temporal counterfactuals.** We introduce CORTEX, the first framework designed to generate counterfactual explanations that are not only valid but also plausible and temporally robust. Unlike prior methods that treat time series as static features or aggregate snapshots, CORTEX explicitly leverages the spatio-temporal nature of the data to ensure coherent, realistic explanations.

**(2) Formalizing temporal robustness as a desideratum.** We define and operationalize the concept of temporal robustness – the requirement that a counterfactual remains in the target class over time – via phase trajectory analysis of model predictions. This extends the standard desiderata of counterfactual explanations (Guidotti, 2022) to dynamic settings, providing a new principle for designing explainers in autoregressive domains.

**(3) Demonstrated reliability in clinical time series.** On EEG-based seizure detection, CORTEX consistently yields explanations that are both more faithful and more realistic than SoTA ($6.32\times$ higher fidelity, and zero implausibility), thanks to its design choices. By grounding explanations in temporally robust, historically observed states, CORTEX enables actionable and reliable recourse in high-stakes applications such as healthcare.

## 2 Preliminaries

Let $\mathcal{G}$ describe the (finite) universe of graphs. Let $G = (X \in \mathbb{R}^{n \times d}, A \in \mathbb{R}^{n \times n}) \in \mathcal{G}$ be a graph with node feature matrix $X$ and (weighted) adjacency matrix $A$, where $n$ is the number of nodes, and $d$ is the feature dimensionality. Let $\Phi : \mathcal{G} \to Y$ be a (already-trained) classification model, where $Y$ is the set of target labels.

Note that the following assumes the reader understands that counterfactual explainability is a post-hoc method applied to $\Phi$, i.e., after $\Phi$ has been trained.

**Definition 1 (Static graph counterfactual)** *Given $\Phi$ and $G \in \mathcal{G}$, a counterfactual for $G$ is found as follows (Jiang et al., 2024; Prado-Romero et al., 2024b):*

$$\underset{\substack{G' \in \mathcal{G} \\ \Phi(G') \neq \Phi(G)}}{\arg\min} \ M(G, G'), \tag{1}$$

*where $M : \mathcal{G} \times \mathcal{G} \to \mathbb{R}_{\geq 0}$ is an optimization metric, generally expressed in dissimilarity terms.*

According to Guidotti (2022), counterfactuals should satisfy validity, minimality, proximity, plausibility, actionability, and causality. Note how Equation (1) satisfies the first three desiderata. While all these properties are important, the last three are seldom treated in other works. We argue that actionability and causality are more related to privacy literature – i.e., actionability ensures that changes only affect modifiable features, and causality requires preserving any known causal feature relationships, usually formalized through structural causal models – and thus not treated in this paper. Contrarily, plausibility is the most overlooked desideratum because it enforces the counterfactuals to be in the distribution of the real data, a phenomenon that learning-based, especially generative, explainers cannot always guarantee (see Ma et al., 2022; Prado-Romero et al., 2024a).

Taking inspiration from Guidotti (2022) and the definition of counterfactual robustness (Jiang et al., 2024), we introduce an additional desideratum, namely *temporal robustness*, which is essential to ensure robustness

of counterfactual explanations in dynamic settings. Temporal robustness requires that counterfactuals remain consistently classified by $\Phi$ over time, i.e., they do not lie in non-robust or transition regions of the feature space. In other words, temporal robustness is needed to guarantee that the generated counterfactual does not immediately revert to an anomalous state as the time series evolves. Without this property, the counterfactual would lose practical relevance, since its validity would not persist under the natural temporal dynamics of the system.

Before we extend the notion of counterfactuals for spatio-temporal graphs, let us define what a time (finite) graph is: i.e., a graph $G_{t_i} = (X_{t_i} \in \mathbb{R}^{n_{t_i} \times d_{t_i}}, A_{t_i} \in \mathbb{R}^{n_{t_i} \times n_{t_i}})$ where $n_{t_i}$ is the number of nodes at time $t_i \in \mathbb{N}$. Now the graph universe $\mathcal{G}$ contains all possible time graphs.

**Definition 2 (Temporally robust graph counterfactual)** *Given $G_{t_i} \in \mathcal{G}$, $\Phi$, and an optimization metric $M : \mathcal{G} \times \mathcal{G} \to \mathbb{R}_{\geq 0}$ (e.g., dissimilarity, temporal distance), a counterfactual for $G_{t_i}$ satisfies validity, minimality, proximity, and temporal robustness if*

$$\underset{\substack{G_{t_j} \in \{\mathcal{G}\}_{<t_i} \\ \Phi(G_{t_j}) \neq \Phi(G_{t_i})}}{\arg\min} \quad M(G_{t_i}, G_{t_j}) \ s.t. \ S_\Phi(G_{t_j}) \leq \epsilon \,, \tag{2}$$

*where $\{\mathcal{G}\}_{<t_i}$ denotes the set of graphs prior to time $t_i$,[2] $S_\Phi : \mathcal{G} \to [0,1]$ is an* instability score *of $\Phi$ on a graph (intended as a lack of temporal robustness), and $\epsilon$ is an upper bound on instability. Smaller values of $S_\Phi$ indicate more temporally robust predictions.*

In words, Definition 2 selects among past candidates that flip $\Phi$'s decision, choosing the closest one that also exhibits sufficiently low predictive volatility. Plausibility can be guaranteed by construction if we restrict the universe of time graphs $\mathcal{G}$ to the observed dataset $\mathcal{D}$, as is typical in search-based explainers (see Section 3). Since the hard constraint in Equation (2) may occasionally yield no feasible solution, we also consider a relaxed formulation in which temporal instability is penalized rather than enforced:

$$\underset{\substack{G_{t_j} \in \{\mathcal{G}\}_{<t_i} \\ \Phi(G_{t_j}) \neq \Phi(G_{t_i})}}{\arg\min} \quad M(G_{t_i}, G_{t_j}) \ + \ S_\Phi(G_{t_j}) \,. \tag{3}$$

While the general task may involve multi-class classification, we focus here on the binary setting, aiming to explain how an anomalous state can be reverted to normality.

## 3 Related Work

Although our work focuses on counterfactual explanations for binary classification of multivariate EEG time series modeled as dynamic graphs, we also include methods for time series and static graphs for completeness.

**Time Series Counterfactual Explainers (TSCEs).** Most counterfactual explanation methods are not designed for time series data. Recent approaches aim to fill this gap. For instance, Ates et al. (2021) generate counterfactuals by replacing entire features with those from distractor samples, optimizing for class change and minimal alteration, though this overlooks temporal dependencies. To address this, Delaney et al. (2021) propose an iterative, nearest-neighbor-based method that balances plausibility and sparsity. Yamaguchi et al. (2024) introduce "deviation intervals", restricting changes to decision-critical time regions for greater temporal focus and interpretability. Despite these advances, most time series methods treat inputs as flat vectors, ignoring their spatio-temporal structure or allowing non-actionable modifications to past inputs.

**Graph Counterfactual Explainers (GCEs).** Most existing counterfactual explainers focus on static graphs. These approaches (Abrate & Bonchi, 2021; Bajaj et al., 2021; Cai et al., 2025; Lucic et al., 2022; Ma et al., 2022; Nguyen et al., 2022; Numeroso & Bacciu, 2021; Prado-Romero et al., 2024a; Prenkaj et al., 2025; Sun et al., 2021; Tan et al., 2022; Wellawatte et al., 2022; Wu et al., 2021) typically perturb the graph minimally (add/remove edges and feature alteration). Search-based techniques (Liu et al., 2021) instead explore the dataset to find optimal counterfactuals based on specific distance metrics (e.g., Graph Edit

---

[2]Note that it is meaningless to produce counterfactuals that succeed the original instance in time.

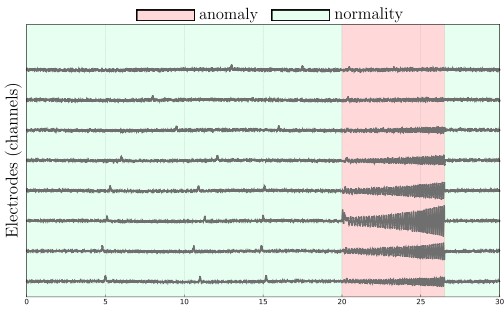

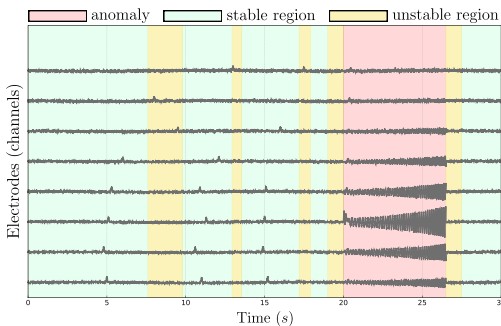

(a) EEG normal phases vs. abnormal ones.

(b) CORTEX identifies, for each anomalous data point at time $t$, robust (i.e., stable) and normal counterfactual regions at time $< t$.

Figure 2: EEG has electrodes (channels) that record electrical impulses from specific brain regions over time. This can be represented as a multivariate time series. An epileptic event in EEG occurs when a population of cortical neurons (in our case, across multiple channels) fires hypersynchronously, leading to sudden spikes, sharp waves, or rhythmic discharges. These can appear interictally (isolated spikes) or ictally (sustained, evolving patterns).

Distance). However, all these methods overlook time. For a detailed review, see the taxonomy in (Prado-Romero et al., 2024b).

**Dynamic GCEs.** Counterfactual explanations in temporal graphs remain underexplored, especially in domains where temporal plausibility is critical, such as epileptic seizure detection. Dynamic graphs are classified into Discrete-Time Dynamic Graphs (DTDGs) and Continuous-Time Dynamic Graphs (CTDGs) (Kazemi et al., 2020). While CTDGs offer fine-grained resolution, their irregular event streams make counterfactual generation particularly challenging. CoDy (Qu et al., 2025) uses a search-based approach with Monte Carlo Tree Search and temporally aware heuristics, but assumes access to continuous event streams, making it unsuitable for DTDGs. In DTDGs, most explainers rely on windowed snapshots. These include explainers based on surrogate models (He et al., 2022), temporal decomposition (Liu et al., 2024), and feature attribution (Fan et al., 2021), but often fail to ensure semantic similarity and plausibility. Generative models like GRACIE (Prenkaj et al., 2024) improve this by modeling class-conditional distributions over snapshots, but still operate at coarse granularity. Generating fine-grained, plausible counterfactuals at the level of individual time steps, without relying on smoothed or aggregated views, remains an open challenge, which we address in this paper.

## 3.1 Challenges and Gaps in SoTA

**TSCEs.** Most TSCEs perturb raw input features or limit changes to learned deviation intervals, often ignoring inter-feature dependencies (Ates et al., 2021; Delaney et al., 2021; Yamaguchi et al., 2024). They also allow direct modifications to past data points, an unrealistic assumption in domains where past states are immutable. Instead, we select temporally robust, structurally coherent past instances as counterfactuals, ensuring both plausibility and validity.

**Static GCEs.** While the explainers in this category capture relational dependencies, they lack temporal awareness and therefore offer limited utility in dynamic settings. We overcome this by treating the system as a sequence of graph snapshots and generating timestamp-level counterfactuals that capture both spatial and temporal structure.

**Dynamic GCEs.** Dynamic GCEs better reflect evolving systems but often target CTDGs or use windowed aggregation in DTDGs, yielding global temporal (Qu et al., 2025) or coarse-grained explanations (Prenkaj et al., 2024). Such approaches are ill-suited to settings that require fine temporal resolution. Our method instead operates at per-timestamp granularity (i.e., DTDG with a window length of 1), identifying robust,

real past observations as counterfactuals and enabling localized, plausible explanations without temporal smoothing.

# 4 Context

Let us contextualize the scenario we are dealing with to ensure the reader understands what normality and anomaly in EEG multivariate time series mean, and why robust counterfactuals are preferred over mere valid ones. EEG signals are collected from multiple electrodes across different brain regions to measure electrical activity. Each channel records the potential difference between a pair of electrodes, and the values collected over time constitute the features of the multivariate time series. We consider a binary classification problem: normal (no epilepsy) versus abnormal (epilepsy) condition (see Figure 2a).[3] Our focus is on providing recourse for anomalous phases – that is, identifying the changes required to terminate hypersynchronous neuronal firing – rather than on the transitions from normal to abnormal states. This is motivated by neuroscience and clinical evidence showing that seizures often self-terminate through specific network- and cellular-scale mechanisms, and that interventions aimed at terminating seizures (e.g., closed-loop stimulation) are a major clinical objective (Kramer et al., 2012; Jiruska et al., 2013; Khambhati et al., 2015; Merelli et al., 2016; Connolly et al., 2024). Counterfactual recourse prescribing how to exit the anomalous state is therefore particularly relevant.

In our setting, this multivariate time series at each time $t_i$ gets transformed into a time graph $G_{t_i}$, where nodes represent channels and the adjacency matrix encodes cross-correlations between channels over a sliding window, following prior work on GNNs for EEG and time series data (Tian & Zhang, 2025; Abadal et al., 2025; Corradini et al., 2026) (see Section B for details). Rather than optimizing for Equation (1) at each $t_i$ in the anomalous phase to simply find another graph $G_{t_j}$ such that $\Phi(G_{t_i}) \neq \Phi(G_{t_j})$, we aim to identify a robust counterfactual that is likely to remain in the normal condition, as defined in Equation (3). Indeed, if we were to select $G_{t_j}$ immediately preceding an anomalous phase, we would obtain a valid counterfactual; however, this approach lacks robustness: advancing time from that state, the system typically returns to the anomalous regime, making $G_{t_j}$ unsuitable in terms of robustness and plausibility (Guidotti, 2022). CORTEX, instead, distributes counterfactuals into "stable" and "unstable" regions, and restricts the search to those unlikely to return to the anomalous condition in the near future, which are considered temporally robust (see Figure 2b) w.r.t. $\Phi$'s outcome.

# 5 Methodology

Here, we introduce our novel search-based explainer, COunterfactual Reasoning for Temporal EXplanations (CORTEX), that accounts for the relationship among feature, time, and space by modeling multivariate EEG series as spatio-temporal graphs. In our setting, each time step $t_i$ of the time series is mapped to a graph $G_{t_i}$ whose nodes correspond to features (here, EEG channels) and edges encode their relationship (here, cross-correlation over a temporal window), following standard practices in the GNN literature for time series data (Corradini et al., 2026). On top of this representation, CORTEX produces counterfactuals – see Section A for an algorithmic overview. Recall that we want to ensure the validity and plausibility of the counterfactuals. Therefore, instead of sampling from a learned latent distribution – see Ma et al., 2022; Prado-Romero et al., 2024a; Prenkaj et al., 2025 among others – we directly search for robust counterfactuals among past observations in the dataset.

**Robust Counterfactual Identification.** Let $\sigma : \mathcal{G} \rightarrow [0, 1]$ be the function that assigns a continuous score to each graph $G_{t_i}$ such that $\Phi(G_{t_i}) = \mathbf{1}[\sigma(G_{t_i}) \geq \theta]$ for a threshold $\theta \in [0, 1]$. For simplicity, we write $\sigma_i = \sigma(G_{t_i})$.

A straightforward way to quantify the temporal robustness of the classification is to track the score $\sigma$ over time. In more detail, given a graph $G_{t_i}$, we calculate $(\sigma_i, v_i)$, where $\sigma_i$ is the sigmoid value associated with $\Phi(G_{t_i})$, and $v_i$ is the sigmoid oscillation velocity calculated over the next $w$ time steps (ensuring it remains

---

[3] Although abnormal events in EEG are rare, making classification potentially unbalanced, we can mitigate this by sampling the channels during normal phases at different frequencies (e.g., 256Hz instead of 512Hz) and downsampling the series from the source, yielding a balanced dataset.

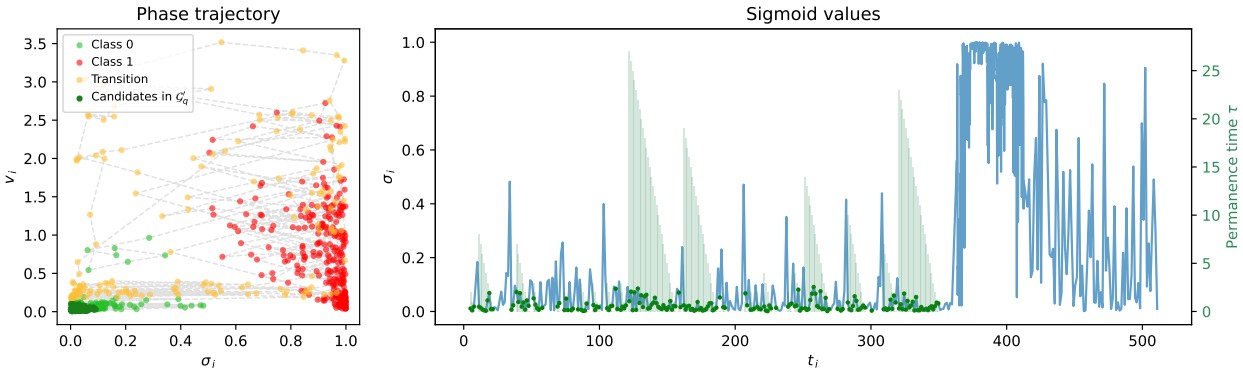

Figure 3: (left) Sigmoid phase trajectory, colored by the oracle classification within the time window of length $w$ (green if all points in the window belong to class 0; red if all belong to class 1; and yellow if the classification changes within the window, indicating transition regions). Darker green points indicate elements in $\mathcal{G}'_q$. (right) Sigmoid values over time, along with points in $\mathcal{G}'_q$ (green points) and their residual permanence times $\tau$. Example from recording `chb03_01`.

within the series length $T$):

$$v_i = \frac{\sum_{j=0}^{w-1} \left| \sigma_{\min(i+j+1,T)} - \sigma_{\min(i+j,T)} \right|}{t_{\min(i+w,T)} - t_i} \, , \tag{4}$$

where the denominator accounts for differences in sampling rates across the classes.[3] High values of $v_i$ correspond to greater amplitudes in the oscillations, which can lead to local instability of the predicted class of $G_{t_i}$ in the immediately succeeding time steps (i.e., lack of temporal robustness). Note that since we are performing a post-hoc analysis, the full time series is already available, and all values used to compute $v_i$ have already been observed. Therefore, this computation does not violate causality, and instability can be reliably measured from the observed signal evolution.

Given a graph $G_{t_i}$ s.t. $\Phi(G_{t_i}) = 1$ (abnormal), we want to find counterfactuals $G_{t_j}$ with a low output-velocity combination. One can map each $(\sigma_i, v_i)$ point into a output-velocity plot (see Figure 3 (left)). Note that the ideal position for $G_{t_j}$ is at $(0,0)$, since the sigmoid value indicates that $\Phi$ classifies $G_{t_j}$ as normal with a high score, and the velocity shows that $G_{t_j}$ has no risk of crossing $\Phi$'s decision boundary in the near future (up to $w$ time steps). Alas, in reality, it is unlikely that all counterfactuals $G_{t_j}$ are in $(0,0)$. Thus, we rank them according to their Euclidean distance $\delta_j = \sqrt{\sigma_j^2 + v_j^2}$ from $(0,0)$. Given the set of valid counterfactual candidates $\mathcal{G}'$ for $G_{t_i}$:

$$\mathcal{G}' = \left\{ (G_{t_j}, \delta_j) \mid \Phi(G_{t_j}) \neq \Phi(G_{t_i}) \ \wedge \ \delta_j = \sqrt{\sigma_j^2 + v_j^2} \right\} , \tag{5}$$

we sort all its elements in ascending order according to $\delta_j$ and select the top $q\%$ to form the subset $\mathcal{G}'_q$:

$$(G_{t_j}, \delta_j) \in \mathcal{G}'_q \iff (G_{t_j}, \delta_j) \in \mathcal{G}' \ \wedge \ \text{rank}_\delta^\uparrow \big( (G_{t_j}, \delta_j), \mathcal{G}' \big) \leq \lceil q \cdot |\mathcal{G}'| \rceil , \tag{6}$$

where $\text{rank}_\delta^\uparrow \big( (G_{t_j}, \delta_j), \mathcal{G}' \big)$ is the 1-based rank of $\delta_j$ in ascending order within $\mathcal{G}'$ (i.e., smallest $\delta$ gets 1), and $\lceil q \cdot |\mathcal{G}'| \rceil$ gives the top $q\%$ cutoff. The parameters $w$ and $q$ in Equations (4) and (6) are selected to achieve the desired level of temporal stability. Specifically, $w$ controls how close candidate points can approach unstable conditions, while $q$ determines the fraction of instances considered robust. Further details on parameter selection and its impact are provided in Section B. Similarly to Equation (2), $\{\mathcal{G}'\}_{<t_i}$ and $\{\mathcal{G}'_q\}_{<t_i}$ denote the sets of valid and plausible counterfactuals for $G_{t_i}$ that occur prior to time $t_i$ in $\mathcal{G}'$ and $\mathcal{G}'_q$, respectively. In what follows, we abuse the notation, and use $G_{t_j} \in \{\mathcal{G}'\}_{<t_i}$ to indicate that $G_{t_j}$ belongs to $\{\mathcal{G}'\}_{<t_i}$ instead of $(G_{t_j}, \delta_j) \in \{\mathcal{G}'\}_{<t_i}$ for readability purposes. The same applies for $\{\mathcal{G}'_q\}_{<t_i}$.[4]

---

[4]If $\{\mathcal{G}'\}_{<t_i} = \varnothing$ – i.e., either no possible counterfactuals due to (rare) poor training of $\Phi$, or every counterfactual is in the future – a valid counterfactual cannot be produced. In such cases, the original instance is returned.

**Multi-Objective Selection Criterion.** Once we obtain $\mathcal{G}_q' \subset \mathcal{G}$, we search for the best counterfactuals by minimizing a metric function $M : \mathcal{G} \times \mathcal{G} \to \mathbb{R}_{\geq 0}$. Given $G_{t_i}$, CORTEX searches over all $G_{t_j} \in \{\mathcal{G}_q'\}_{<t_i}$ and optimizes $M$ containing three components: (1) the dissimilarity between $G_{t_i}$ and $G_{t_j}$, (2) their temporal distance, and (3) the robustness of $G_{t_j}$.

*(1) Dissimilarity Metric.* We define $M_\alpha$ as the Frobenius norm of the difference between the product of the Laplacians of $G_{t_i}$ and $G_{t_j}$ and their node feature matrices:

$$M_\alpha(G_{t_i}, G_{t_j}) = \left\| L_{t_i} X_{t_i} - L_{t_j} X_{t_j} \right\|_F \, , \tag{7}$$

where $L_{t_i} \in \mathbb{R}^{|n_{t_i}| \times |n_{t_i}|}$ is the normalized Laplacian matrix of graph $G_{t_i}$ defined as $L_{t_i} = I - D_{t_i}^{-1/2} A_{t_i} D_{t_i}^{-1/2}$, with $D_{t_i} \in \mathbb{N}^{n_{t_i} \times n_{t_i}}$ the diagonal degree matrix of $G_{t_i}$, and $I \in \mathbb{R}^{n_{t_i} \times n_{t_i}}$ the identity matrix. The attentive reader would notice that Equation (7) is defined only when $n_{t_i} = n_{t_j}$, and is meaningful only if the nodes are considered in the same order. This condition holds in our scenario because each node consistently represents the same feature across the multivariate time series (see also Section 6.2). Note also that $M_\alpha$ does not rely on $\Phi$ for its computation; this is because distances between oracle embeddings would reflect differences in class representations rather than single graph dissimilarities. Indeed, $\Phi$ tends to map instances of the same class close together in the embedding space, even if the graphs differ significantly. As a result, oracle-based distances would be biased towards class semantics and overlook structural graph differences.

*(2) Temporal Distance.* We also want to favor counterfactuals in $\{\mathcal{G}_q'\}_{<t_i}$ that are temporally close to the original instance. Thus, we define $M_\beta$ as the quadratic difference between $t_i$ and $t_j$:

$$M_\beta(G_{t_i}, G_{t_j}) = (t_i - t_j)^2 \, . \tag{8}$$

We chose a quadratic form to heavily penalize graphs that are further apart in time. However, a study of other penalization functions (e.g., linear) is left for future work.

*(3) Temporal Robustness.* For each $G_{t_j} \in \mathcal{G}_q'$, we define the residual permanence time $\tau_j$ as the number of consecutive time steps during which the sequence remains in $\mathcal{G}_q'$. Formally, let the permanence time $\tau_j$ for $G_{t_j} \in \mathcal{G}_q'$ be:

$$\tau_j = \max\{h \in \mathbb{N} \mid G_{t_{j+h}} \in \mathcal{G}_q'\} \, . \tag{9}$$

Thus, $\tau_j$ counts the length of the maximal contiguous block $G_{t_{j+1}}, G_{t_{j+2}}, \ldots$ that lies entirely in $\mathcal{G}_q'$. We then normalize $\tau_j$ into an instability score $M_\gamma$, mapping the largest permanence to 0 (most robust) and the smallest permanence to 1 (least robust), with intermediate values scaled linearly between 0 and 1. Formally, for all $G_{t_j}$ with (finite) permanence time $\tau_j$ we define

$$M_\gamma(G_{t_j}) = \frac{\tau_{\max} - \tau_j}{\tau_{\max} - \tau_{\min}} \, , \tag{10}$$

where $\tau_{\max} = \max\{\tau_j\}$ and $\tau_{\min} = \min\{\tau_j\} \; \forall G_{t_j} \in \mathcal{G}_q'$ (by construction, $\tau_{\min} = 0$). If $\tau_{\max} = 0$, the set $\mathcal{G}_q'$ only contains isolated robust counterfactual candidates. In this case, to avoid division by zero in Equation (10), we set $M_\gamma(G_j) \equiv 1$. Note that $M_\gamma$ is computed for all $G_{t_j}$ and depends on the entire set $\mathcal{G}_q'$ and on the hyperparameters $w$ and $q$. Note also how Equation (10) subsumes the role of $S_\Phi$ in Equation (3) since it associates every graph with an instability score over the top-$q^{\text{th}}$ most robust percentile of counterfactuals in $\mathcal{G}_q'$. We illustrate the permanence time with the shaded areas in Figure 3 (right).

CORTEX optimizes Equation (11) to find a counterfactual given a graph $G_{t_i}$, where each term is first normalized in $[0, 1]$ across all candidates in $\{\mathcal{G}_q'\}_{<t_i}$ to balance their contributions to the overall metric:

$$\underset{G_{t_j} \in \{\mathcal{G}_q'\}_{<t_i}}{\arg\min} \underbrace{M_\alpha(G_{t_i}, G_{t_j}) + M_\beta(G_{t_i}, G_{t_j})}_{M \text{ in Eq.(3)}} + \underbrace{M_\gamma(G_{t_j})}_{S_\Phi \text{ in Eq.(3)}} \, . \tag{11}$$

Note that one can also produce the top-$k$ counterfactuals rather than the top-1 as described above.

**Satisfied Counterfactuality Properties.** CORTEX satisfies all desiderata in (Guidotti, 2022) (besides actionability and causality, as justified in Section 2). Validity and plausibility hold since the search for

counterfactuals is done over $\{\mathcal{G}'_q\}_{<t_i}$ which, by construction, should contain valid and plausible ones – see Equations (5) and (6). Proximity is enforced via $M_\alpha$ that accounts for both topology and node features. Finally, we guarantee temporal robustness by selecting robust candidates in $\{\mathcal{G}'_q\}_{<t_i}$ (see Figure 3 (left)) and further reinforce it through $M_\gamma$.

Notice that CORTEX operates on previously detected anomalies, analyzing their temporal evolution *a posteriori* to highlight structural differences and support clinicians in guiding recovery towards normal EEG states. Although implemented offline in our experiments, the method can be adapted to streaming data: as long as anomalous segments are flagged by a detector and past normal data is available, CORTEX can provide explanations and guide transitions towards the normal condition. The only required modification concerns the definition of $v_i$ in Equation (4), which currently relies on future time steps. In a streaming setting, $v_i$ can be redefined using past time steps instead, or replaced by a proxy for temporal instability, potentially based on predictions of future behavior. All these aspects related to streaming data will be investigated in future work.

## 6 Experiments

### 6.1 Datasets

*PhysioNet CHB-MIT Scalp EEG* (Shoeb, 2010) contains EEG recordings from 22 pediatric patients sampled at 256 Hz. *PhysioNet Siena Scalp EEG* (Detti, 2020) has 14 adult patient EEG recordings sampled at 512 Hz. Since patients differ in their EEG characteristics, both the oracle training and the counterfactual generation are performed on a per-patient basis to maximize the accuracy and generalizability of explanations. For this reason, we selected a subset of patients with the highest number of recordings and seizures: patients `chb01`, `chb03`, and `chb10` from the first dataset, and patients `PN00`, `PN06`, and `PN14` from the second. For each patient, we use all recordings with seizure events that share the same set of EEG channels to ensure data consistency, resulting in a total of 31 recordings to be studied (recording IDs can be found in Tables 1 and 2; each ID is in the form `patientID_recordingID`, where the string before the underscore indicates the patient and the number after indicates the recording). This selection provides richer intra-patient data and ensures that our model is evaluated on consistent, high-quality data, thereby improving the accuracy and generalizability of patient-specific explanations. Each recording contains time-varying graphs, where nodes correspond to EEG channels and edges encode their cross-correlations. The graphs reflect the spatial configuration of the electrodes and the temporal dynamics of their interactions. We point the reader to Section B for a detailed description of how to build these time graphs from the patient recordings.

### 6.2 Experimental Setup

**Oracle training.** For each patient, we train a 2-layer GCN end-to-end on all their EEG recordings to predict the label of each graph $G_{t_i}$ at its time $t_i$. We are aware that various GNN architectures, including spatio-temporal models (Corradini et al., 2026), can be used. However, we refrain from committing to a specific design, as CORTEX is a post-hoc explainer and a full architectural comparison of the oracle $\Phi$ is outside the scope of this work. Once trained, $\Phi$ acts as a black-box oracle, providing predictions throughout the counterfactual generation pipeline.

**Baselines.** We compare CORTEX[5] with two heuristic-based explainers – i.e., OBS and DDBS (Abrate & Bonchi, 2021) – and a hybrid-based (search+heurstic) explainer GNN-MOExp (Liu et al., 2021). DDBS finds counterfactuals by filtering edges according to their probability of appearing in graphs of a given class. In our experiments, we adapt DDBS to compute these probabilities solely from past instances, thereby preventing the use of illegitimate future information in time series settings.

We noticed that learning-based explainers (e.g., Ma et al., 2022) are unsuitable in our scenario because most recordings described above have a single anomalous phase. Since learning-based explainers require a mandatory training process, splitting the time series into train-test portions might yield splits that do not preserve temporal dependencies (i.e., training data in the future, test data in the past). In this way,

---

[5]Source code: `https://github.com/MartinaZan/cortex.git`.

during training, these explainers would see the future beforehand, and they would not guarantee to produce a counterfactual that is in the past (see the conditions under $\arg\min$ in Equations (2) and (3)). Even a leave-one-subject-out approach is not recommended in this context, given the high variability across patients. In such a scenario, the explainer would primarily learn general patterns across patients and recordings. In practice, the resulting explanations would be less reliable and less informative, as they reflect population-level patterns rather than the patient-specific EEG characteristics that should underpin them. Hence, we exclude them from our analysis. We also exclude TSCEs from our comparison, as they generate explanations by modifying past instances, making their results not directly comparable with CORTEX. In future work, we plan to adapt all these explainers to time graphs. Recall that CORTEX belongs to the class of DTDG dynamic explainers, and the only work in the literature is GRACIE (Prenkaj et al., 2024). However, GRACIE requires creating two VGAE "experts" that represent the two EEG classes at time $t_0$. Because, at $t_0$, each recording has only one graph, GRACIE fails to learn these representations. Hence, it is a meaningless competitor in our scenario.

## 6.3 Results and Analysis

**CORTEX outperforms SoTA in terms of validity (3.73× higher) and fidelity (6.32× higher).** Tables 1 and 2 report average validity and fidelity of CORTEX and SoTA methods across all 31 EEG recordings, based on the corresponding patient-specific oracle $\Phi$. CORTEX consistently achieves the highest scores, outperforming all SoTA models across the board. This performance is the result of a key design choice: counterfactuals are generated by searching only among valid and temporally robust past instances. Errors occur only when explaining instances misclassified by the oracle (i.e., when explaining false negatives) for which CORTEX has no valid candidate in $\{\mathcal{G}'_q\}_{<t_i}$. Contrarily, SoTA models primarily focus on perturbing the original instance while minimizing dissimilarity, but this often comes at the cost of validity.

Table 1: CORTEX vs. SoTA in terms of validity (↑). Best results in **bold**, second-best underlined. CORTEX has a 3.73× higher validity than the second-best performing method across the board.

| | chb01_03 | chb01_04 | chb01_15 | chb01_16 | chb01_18 | chb01_21 | chb01_26 | chb03_01 | chb03_02 | chb03_03 | chb03_04 | chb03_34 | chb03_35 | chb03_36 | chb10_12 |
|---|---|---|---|---|---|---|---|---|---|---|---|---|---|---|---|
| OBS | 0.1037 | 0.3255 | 0.1724 | 0.4848 | 0.5325 | 0.2076 | 0.4122 | 0.3227 | 0.1629 | 0.2294 | 0.3008 | 0.3501 | 0.2365 | 0.2100 | 0.4468 |
| DDBS | 0.0469 | 0.1693 | 0.0517 | 0.3384 | 0.4431 | 0.0506 | 0.1888 | 0.1493 | 0.0451 | 0.1322 | 0.1303 | 0.1289 | 0.0468 | 0.0787 | 0.1835 |
| GNN-MOExp | 0.0617 | 0.1745 | 0.0690 | 0.1433 | 0.1341 | 0.0532 | 0.0798 | 0.1493 | 0.0902 | 0.1596 | 0.2105 | 0.2577 | 0.1034 | 0.1391 | 0.0718 |
| CORTEX | **1.0000** | **1.0000** | **1.0000** | **1.0000** | **1.0000** | **1.0000** | **1.0000** | **1.0000** | **1.0000** | **1.0000** | **1.0000** | **1.0000** | **1.0000** | **1.0000** | **1.0000** |

| | chb10_20 | chb10_27 | chb10_30 | chb10_38 | chb10_89 | PN00_1 | PN00_2 | PN00_4 | PN00_5 | PN06_2 | PN06_3 | PN06_4 | PN06_5 | PN14_1 | PN14_2 | PN14_4 |
|---|---|---|---|---|---|---|---|---|---|---|---|---|---|---|---|---|
| OBS | 0.1969 | 0.1028 | 0.4083 | 0.1813 | 0.3307 | 0.9599 | 0.5417 | 0.6566 | 0.2610 | 0.5257 | 0.5923 | 0.3764 | 0.7695 | 0.5311 | 0.1610 | 0.0471 |
| DDBS | 0.0787 | 0.0401 | 0.1705 | 0.0879 | 0.1601 | 0.9091 | 0.4083 | 0.5412 | 0.1951 | 0.4562 | 0.4962 | 0.2802 | 0.7366 | 0.3672 | 0.1384 | 0.0443 |
| GNN-MOExp | 0.0997 | 0.0952 | 0.1163 | 0.1071 | 0.2021 | 0.9465 | 0.9028 | 0.8819 | 0.4176 | 0.4350 | 0.5231 | 0.5604 | 0.3745 | 0.6045 | 0.2571 | 0.0886 |
| CORTEX | **1.0000** | **1.0000** | **1.0000** | **1.0000** | **1.0000** | **1.0000** | **1.0000** | **1.0000** | **1.0000** | **1.0000** | **1.0000** | **1.0000** | **1.0000** | **1.0000** | **1.0000** | **1.0000** |

Table 2: CORTEX vs. SoTA in terms of fidelity (↑). Best results in **bold**, second-best underlined. CORTEX has a 6.32× higher fidelity than the second-best performing method across the board.

| | chb01_03 | chb01_04 | chb01_15 | chb01_16 | chb01_18 | chb01_21 | chb01_26 | chb03_01 | chb03_02 | chb03_03 | chb03_04 | chb03_34 | chb03_35 | chb03_36 | chb10_12 |
|---|---|---|---|---|---|---|---|---|---|---|---|---|---|---|---|---|
| OBS | 0.0439 | 0.1326 | 0.0719 | 0.1550 | 0.1101 | 0.0884 | 0.1731 | 0.1381 | 0.0747 | 0.0818 | 0.1296 | 0.1415 | 0.0964 | 0.0828 | 0.1933 |
| DDBS | 0.0179 | 0.0705 | 0.0315 | 0.1070 | 0.0835 | 0.0267 | 0.0879 | 0.0716 | 0.0481 | 0.0585 | 0.0490 | 0.0229 | 0.0284 | 0.0785 | 0.0785 |
| GNN-MOExp | 0.0499 | 0.0951 | 0.0455 | -0.0254 | -0.1567 | 0.0300 | 0.0233 | 0.0901 | 0.0877 | 0.0964 | 0.1571 | 0.1947 | 0.0631 | 0.0960 | 0.0030 |
| CORTEX | **0.9258** | **0.8910** | **0.4750** | **0.7263** | **0.6106** | **0.9259** | **0.8700** | **0.8953** | **0.9372** | **0.9159** | **0.9454** | **0.9147** | **0.9178** | **0.9261** | **0.8672** |

| | chb10_20 | chb10_27 | chb10_30 | chb10_38 | chb10_89 | PN00_1 | PN00_2 | PN00_4 | PN00_5 | PN06_2 | PN06_3 | PN06_4 | PN06_5 | PN14_1 | PN14_2 | PN14_4 |
|---|---|---|---|---|---|---|---|---|---|---|---|---|---|---|---|---|
| OBS | 0.0766 | 0.0449 | 0.1853 | 0.0966 | 0.1412 | 0.4405 | 0.1918 | 0.2346 | 0.0998 | 0.1095 | 0.1001 | 0.0862 | 0.1684 | 0.1801 | 0.0489 | 0.0143 |
| DDBS | 0.0314 | 0.0186 | 0.0824 | 0.0500 | 0.0706 | 0.4080 | 0.1323 | 0.1781 | 0.0698 | 0.0877 | 0.0747 | 0.0562 | 0.1522 | 0.1270 | 0.0433 | 0.0121 |
| GNN-MOExp | 0.0615 | 0.0748 | 0.0580 | 0.0828 | 0.1310 | **0.8729** | **0.7583** | **0.7435** | 0.3203 | 0.1724 | 0.1698 | 0.2953 | 0.0485 | 0.3868 | 0.2107 | 0.0683 |
| CORTEX | **0.9285** | **0.9067** | **0.9057** | **0.7965** | **0.8740** | 0.8202 | 0.6686 | 0.6095 | **0.7912** | **0.5546** | **0.3643** | **0.4710** | **0.5368** | **0.7235** | **0.7713** | **0.8753** |

To provide a concise overview, Table 3 presents the mean and standard deviation of validity and fidelity for each method across subjects. Statistical significance is assessed using a paired two-sample $t$-test, comparing CORTEX with each baseline. CORTEX consistently achieves the highest validity and fidelity scores. All paired comparisons with CORTEX are statistically significant ($p < 0.001$), indicating that the performance differences are robust across subjects.

Table 3: Summary of evaluation metrics. For each model, the mean $\pm$ standard deviation is reported for validity and fidelity, along with the paired $t$-test statistic comparing each method with CORTEX ($p$-values in parentheses).

| Model | Validity | $t$-test ($p$-value) | Fidelity | $t$-test ($p$-value) |
|---|---|---|---|---|
| OBS | $0.3594 \pm 0.2059$ | $17.042$ ($< 0.001$***) | $0.1268 \pm 0.0766$ | $19.386$ ($< 0.001$***) |
| DDBS | $0.2353 \pm 0.2132$ | $19.642$ ($< 0.001$***) | $0.0774 \pm 0.0724$ | $19.980$ ($< 0.001$***) |
| GNN-MOExp | $0.2745 \pm 0.2576$ | $15.426$ ($< 0.001$***) | $0.1829 \pm 0.2185$ | $10.771$ ($< 0.001$***) |
| CORTEX | $\mathbf{1.0000 \pm 0.0000}$ | — | $\mathbf{0.7852 \pm 0.1640}$ | — |

**Including $M_\beta$ and $M_\gamma$ in Equation (11) has little effect on dissimilarity $M_\alpha$.** Table 4 shows the impact of $\{\mathcal{G}'\}_{<t_i}$ vs. $\{\mathcal{G}'_q\}_{<t_i}$, as well as the temporal distance and robustness terms of Equation (11) on the top-5 counterfactuals on the recording `chb03_01`. Note that for explainers performing searches over $\{\mathcal{G}'\}_{<t_i}$, the term $M_\gamma$ is not defined.

Table 4: Study of variants of Equation (11) and the search space. We show mean and standard deviation of $M_\alpha$, $\sqrt{M_\beta}$, and $M_\gamma$ computed over the top-5 counterfactuals for the recording `chb03_01` (oracle accuracy: 0.9336). The components of all optimization variants are normalized in $[0, 1]$ as described in Equation (11). All values are computed only on the explanations for the correctly classified instances (i.e., true positives). Last row is CORTEX.

| Variant | Search space | $M_\alpha$ ($\downarrow$) | $\sqrt{M_\beta}$ ($\downarrow$) | $M_\gamma$ ($\downarrow$) |
|---|---|---|---|---|
| $M = M_\alpha + M_\beta$ | $\{\mathcal{G}'_q\}_{<t_i}$ | $0.439 \pm 0.004$ | $61.787 \pm 16.309$ | $0.714 \pm 0.203$ |
| $M = M_\alpha + M_\beta$ | $\{\mathcal{G}'\}_{<t_i}$ | $0.419 \pm 0.011$ | $39.282 \pm 10.132$ | — |
| $M = M_\alpha$ | $\{\mathcal{G}'_q\}_{<t_i}$ | $0.433 \pm 0.003$ | $189.886 \pm 86.495$ | $0.782 \pm 0.187$ |
| $M = M_\alpha$ | $\{\mathcal{G}'\}_{<t_i}$ | $0.430 \pm 0.004$ | $182.502 \pm 88.909$ | — |
| $M = M_\alpha + M_\gamma$ | $\{\mathcal{G}'_q\}_{<t_i}$ | $0.451 \pm 0.008$ | $179.553 \pm 91.993$ | $0.150 \pm 0.062$ |
| $M = M_\alpha + M_\beta + M_\gamma$ | $\{\mathcal{G}'_q\}_{<t_i}$ | $0.451 \pm 0.006$ | $66.560 \pm 3.980$ | $0.230 \pm 0.065$ |

The values of $M_\alpha$ remain consistent across the board. $M_\beta$ makes CORTEX choose counterfactuals closer in time to the original graph, with a small standard deviation (see $\sqrt{M_\beta}$), indicating that the counterfactuals are more similar to each other w.r.t. the underlying dynamic conditions. Notice that $M_\beta$ and $M_\gamma$ are specular optimization metrics (see rows 5 and 6). $M_\gamma$ prefers counterfactuals that stay longer in "stable" regions – i.e., the left-most data point in a stable region is preferred, but it is also further away from the original instance than the right-most one as shown in Figure 3 (right). $M_\beta$ penalizes this temporal distance. However, when both are included (last row), there is a staggering trade-off where $M_\beta$ improves by 62.93% whereas $M_\gamma$ is just increased by 0.08 points. Notice also that when both these metrics are included, the standard deviation of $M_\beta$ is the lowest among all variants. Finally, optimizing for $M_\gamma$ also yields the largest gain (compare rows 1 and 3 with 5 and 6). As expected from Equation (3) and Section 5, the robustness of counterfactuals clearly ameliorates their quality ($-67.8\%$ in $M_\gamma$) at only a small increased temporal distance ($+7.72\%$ in $\sqrt{M_\beta}$) and increased dissimilarity ($+2.73\%$ in $M_\alpha$) – compare row 1 with last.

### 6.4 Discussion: Plausibility vs. Dissimilarity

By design, CORTEX achieves zero implausibility by restricting the counterfactual search to robust instances within the dataset, effectively producing counterfactuals that have already been observed. Contrarily, OBS and DDBS generate explanations by modifying the original graph. As a consequence, the counterfactuals remain close to the original instance (low dissimilarity) rather than truly reflecting the characteristics of graphs that $\Phi$ would classify as normal (resulting in high implausibility). GNN-MOExp does not enforce the same nodes between the original and counterfactual graphs, resulting in an undefined implausibility score (see Equation (15)).

Notably, when implausibility is high, dissimilarity becomes irrelevant, since such counterfactuals are inherently unsuitable regardless of their proximity to the original graph. In other words, we argue that plausibility is a necessary condition: *only counterfactuals with low implausibility should be considered, and only then, dissimilarity becomes a meaningful criterion for further selection.* Even if OBS and DDBS show lower dissimilarity scores due to generating counterfactuals with identical node features and binary adjacency matrices, this alone does not guarantee explanation quality if plausibility is not satisfied. See Section D for details on the implausibility-dissimilarity relationship among the explainers.

## 7  Generalizability

Although the current study focuses on seizure detection, the methodology is inherently domain-agnostic and has strong potential for application to other physiological signals, as well as to time series in domains such as financial data or industrial sensor monitoring. For instance, in an industrial monitoring scenario, multiple interconnected measurements could be represented as a spatio-temporal graph, and temporally robust counterfactuals could highlight minimal changes that characterize anomalous conditions, supporting predictive maintenance and failure prevention.

To use CORTEX effectively, we recommend datasets with the following characteristics. First, the series should be homogeneous and naturally suited to a graph representation, which makes the generated counterfactual graphs easier to interpret. In this regard, the literature on spatio-temporal GNNs for time series data provides useful guidance. Second, the number of features (i.e., nodes) should remain manageable to keep explanations interpretable. We recommend an ideal range of 15-30 features to ensure the resulting counterfactual graph remains readable. Finally, datasets containing sequential anomalies are preferred, as these are the primary focus of our analysis.

Once a dataset has been identified, the key adaptation is to calibrate parameters for identifying stable candidates in Equations (4) and (6) to domain-specific needs, as shown in Section B. While this calibration is currently manual and tailored to the EEG-based seizure detection problem considered in this study, developing more general automated strategies for parameter selection will be the focus of future work and an important step towards realizing the full cross-domain potential of CORTEX beyond this specific setting.

More generally, the CORTEX methodology, and in particular the principle of temporal robustness, can be applied beyond graph-based representations once an alternative dissimilarity metric and a method to capture inter-dependencies are defined. In this case, the resulting counterfactual would no longer take the form of a graph, but it would preserve the desired properties of plausibility and temporal robustness. We consider CORTEX a first approach for generating plausible, temporally robust counterfactuals, with the potential for broad applicability across datasets and domains.

## 8  Conclusion

We introduced CORTEX, a novel search-based explainer that finds reliable, robust counterfactuals for multivariate EEG series represented as spatio-temporal graphs. By moving beyond the conventional focus on minimizing feature perturbations, CORTEX explicitly addresses the critical needs of dynamic, high-stakes domains by proposing temporal robustness – i.e., ensuring the suggested recourse remains valid over subsequent time steps – as a new desideratum for counterfactuals. Evaluated on EEG-based seizure detection,

CORTEX significantly outperformed the existing SoTA, achieving $3.73\times$ higher validity and a $6.32\times$ higher fidelity, while crucially maintaining zero implausibility.

Building upon these foundations, future work directions include exploring alternative penalization functions for the temporal distance metric to better calibrate the importance of time, and generalizing the method to handle a wider variety of dynamic graph structures. Extending the approach to multi-class classification is another important direction; in this case, it is necessary to determine whether counterfactual explanations aim to transition from a given class to any other class or to a specific target class, and to adapt the analysis of temporal dynamics and stability along the softmax phase trajectory accordingly. Extending CORTEX to streaming scenarios is also an interesting avenue: in this context, $v_i$ could be computed using only past time steps, or replaced with a proxy for temporal instability based on predictions of future behavior. Finally, integrating explicit notions of actionability and causality will also be essential to further enhance CORTEX's clinical utility and trustworthiness for expert practitioners.

### Author Contributions

**Martina Zannotti:** Conceptualization, Methodology, Investigation, Software, Writing - Original Draft, Writing - Review & Editing. **Bardh Prenkaj:** Conceptualization, Methodology, Writing - Original Draft, Writing - Review & Editing, Supervision. **Marco Piangerelli:** Conceptualization, Methodology, Writing - Original Draft, Writing - Review & Editing, Supervision. **Flavio Corradini:** Project administration, Funding acquisition, Supervision. **Gjergji Kasneci:** Methodology, Writing - Review & Editing, Supervision.

### Acknowledgments

This publication was produced while Martina Zannotti was attending the Doctoral Research Program of National Interest in "Blockchain and Distributed Ledger Technology", with the support of a scholarship co-financed by the European Union NGEU - NRRP, Mission 4, Component 2 (CUP J11J23001450006) with the contribution of the company Syeew s.r.l..

Bardh Prenkaj was partially supported by the Friedrich Schiedel Fellowship, hosted by the TUM School of Social Sciences and Technology and the TUM Think Tank at the Munich School of Politics and Public Policy.

This work has been partially funded by the European Union - NextGenerationEU, Mission 4, Component 2, under the Italian Ministry of University and Research (MUR) National Innovation Ecosystem grant ECS00000041 - VITALITY - CUPJ13C22000430001.

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

# A   Algorithmic Overview

For clarity and completeness, we show the pseudo-code of CORTEX in Algorithm 1.[6]

---

**Algorithm 1** CORTEX

---

1: **Input:** $\mathcal{G} = \{G_{t_1}, G_{t_2}, \ldots, G_{t_T}\}$; pre-trained oracle $\Phi : \mathcal{G} \rightarrow Y$; threshold $0 \leq \theta \leq 1$; $w > 0$, $q > 0$; number of counterfactuals $k$
2: **Output:** top-$k$ counterfactuals for each $G_{t_i}$ s.t. $\Phi(G_{t_i}) = 1$
3:
4: *– Robust candidates identification –*
5: Get $\sigma_i = \sigma(G_{t_i})$ s.t. $\mathbf{1}[\sigma(G_{t_i}) \geq \theta] = \Phi(G_{t_i})$ $\forall G_{t_i} \in \mathcal{G}$
6: **for** $G_{t_i} \in \mathcal{G}$ **do**
7: $\quad v_i \leftarrow \dfrac{\sum_{j=0}^{w-1} \left| \sigma_{\min(i+j+1,T)} - \sigma_{\min(i+j,T)} \right|}{t_{\min(i+w,T)} - t_i}$, as in Equation (4)
8: $\quad \delta_i \leftarrow \sqrt{\sigma_i{}^2 + v_i{}^2}$
9: **end for**
10: Compute $\mathcal{G}'$ as in Equation (5)
11: Compute $\mathcal{G}'_q$ as in Equation (6)
12:
13: *– Optimization to compute $M_\gamma$ only once $\forall G_{t_j} \in \mathcal{G}'_q$ –*
14: $\mathcal{T} \leftarrow [\,]$     *Initialize empty list*
15: **for** $G_{t_j} \in \mathcal{G}'_q$ **do**
16: $\quad$ Compute $\tau_j$ as in Equation (9)
17: $\quad \mathcal{T}[j] \leftarrow \tau_j$
18: **end for**
19: $\min_\tau \leftarrow \min(\mathcal{T})$, $\max_\tau \leftarrow \max(\mathcal{T})$
20: $\gamma \leftarrow \dfrac{\max_\tau - \mathcal{T}}{\max_\tau - \min_\tau}$     *Now $\gamma$ is an array that has all taus normalized as in Equation (10)*
21:
22: *– Multi-objective selection –*
23: $cf \leftarrow \text{dict}()$     *Empty dictionary that is going to contain the top-k counterfactuals for each $G_{t_i}$*
24: $\alpha, \beta \leftarrow [[+\infty * |\mathcal{G}|] * |\mathcal{G}'_q|]$     *Initialize a matrix of dimensionality $|\mathcal{G}| \times |\mathcal{G}'_q|$ with elements equal to $+\infty$*
25: **for** $G_{t_i} \in \mathcal{G}$ s.t. $\Phi(G_{t_i}) = 1$ **do**
26: $\quad M \leftarrow +\infty * [|\mathcal{G}'_q|]$     *List of length $|\{\mathcal{G}'_q\}_{<t_i}|$ with all elements equal to $+\infty$*
27: $\quad$ **for** $G_{t_j} \in \{\mathcal{G}'_q\}_{<t_i}$ **do**
28: $\quad\quad \alpha[i,j] \leftarrow M_\alpha(G_{t_i}, G_{t_j})$
29: $\quad\quad \beta[i,j] \leftarrow M_\beta(G_{t_i}, G_{t_j})$
30: $\quad$ **end for**
31: $\quad$ Normalize $\alpha[i,:]$ and $\beta[i,:]$
32: $\quad M \leftarrow \alpha[i,:] + \beta[i,:] + \gamma$
33: $\quad \text{sort}(M, \text{ascending})$
34: $\quad cf[G_{t_i}] \leftarrow M[:k]$     *Take the first k elements, and return a list thereof*
35: **end for**
36: **return** $cf$

---

We assume that all initializations are done in constant time and do not contribute to the overall time complexity.

The identification of robust candidates (lines 5-11) is performed as a preprocessing step after the oracle has been trained. This step is executed only once, regardless of the number of anomalous instances to be explained. Let $P = |\mathcal{G}|$, $R = |\mathcal{G}'|$, $Q = |\mathcal{G}'_q|$, and the time complexity of querying $\Phi$ for the outcome of graphs

---

[6]The pipeline is built upon the GRETEL framework (Prado-Romero & Stilo, 2022). The source code for CORTEX is available at `https://github.com/MartinaZan/cortex.git`.

equal to $\mathcal{O}(1)$ (i.e., line 5 costs $\mathcal{O}(1)$). We also consider the worst-case scenario in which $|\{\mathcal{G}'\}_{<t_i}| = |\mathcal{G}'| = R$, $|\{\mathcal{G}'_q\}_{<t_i}| = |\mathcal{G}'_q| = Q$.

The identification of robust candidates costs $\mathcal{O}(wP) = \mathcal{O}(P)$ since $w$ is a constant. The computation of $\mathcal{G}'$ is $\mathcal{O}(1)$ since we already have calculated $v_i$ and $\delta_i$ for each $G_{t_i}$ in lines 6-9. The computation of $\mathcal{G}'_q$ costs $\mathcal{O}(R \log R)$ since the $\text{rank}_\delta^\uparrow$ function in Equation (6) can be seen as a sorting algorithm over $\mathcal{G}'$ in ascending order and then selecting the first $q$ elements.

Before optimizing the multi-objective of Equation (11), we notice that, regardless of the input graph to be explained, the graphs in $\mathcal{G}'_q$ can have their $M_\gamma$ computed *a priori* and only once (lines 14-20). To compute $\tau_j$ we can scan left-to-right and detect maximal contiguous runs of indices where $G_t \in \mathcal{G}'_q$. For a run of length $h$ that starts at index $s$ (i.e., indices $s, s+1, \ldots s+h-1$ are in the set and $s+h$ is not or sequence ends), the permanence times are: $\tau_s = h-1, \tau_{s+1} = h-2, \ldots, \tau_{s+h-1} = 0$. Filling these values costs $\mathcal{O}(h)$ for a run. Summed over runs this is $\mathcal{O}(Q)$ assuming that the membership $G_t \notin \mathcal{G}'_q$ is negligible. Hence, line 16 costs $\mathcal{O}(Q)$. Finding $\tau_{\min}$ and $\tau_{\max}$ costs $O(Q)$, and computing $\gamma$ (line 20) costs $O(Q)$ since one can imagine this line to be an iteration over $\mathcal{G}'_q$ again to compute the gammas. Therefore, lines 14-20 cost $\mathcal{O}(Q + Q + Q) = \mathcal{O}(Q)$.

Let us now see how much computing $M_\alpha$ and $M_\beta$ costs. Assuming that our graphs are memorized sparsely, computing their Laplacians costs $\mathcal{O}(e)$ where $e$ is the number of edges. Then, the matrix multiplication between the Laplacian and the node feature vectors costs (naively) $\mathcal{O}(n^3)$ where $n$ is the number of nodes. Hence the cost of $M_\alpha$ (line 28) is $\mathcal{O}(n^3 + e) = \mathcal{O}(n^3)$. $M_\beta$ (line 29) is just a simple difference between the time indices of two graphs and that can be done in $\mathcal{O}(1)$. Now, lines 27-30 cost $\mathcal{O}(n^3 Q)$. Line 31 costs $\mathcal{O}(Q)$ since it is just a simple min-max normalization and can be done similarly to lines 19-20. Line 32 is negligible. Line 33 costs $\mathcal{O}(Q \log Q)$ since the list $M$ contains $Q$ elements. Hence, the entire multi-objective selection procedure has a time complexity of $\mathcal{O}(P \cdot (n^3 Q + Q + Q \log Q)) = \mathcal{O}(n^3 PQ + PQ + PQ \log Q) = \mathcal{O}(n^3 PQ + PQ \log Q)$.

The total time complexity of CORTEX is $\mathcal{O}(P + R \log R + Q + n^3 PQ + PQ \log Q)$.

Since $Q < R < P$ (and assuming $Q \geq 2$) we have $\log Q \geq 1$. In this case, the lone $P$ is dominated by $PQ \log Q$, and the term $Q$ is also dominated by $PQ \log Q$ because $P > Q$. Thus:

$$\mathcal{O}(P + R \log R + Q + n^3 PQ + PQ \log Q) = \mathcal{O}(PQ \log Q + n^3 PQ + R \log R).$$

Factor $PQ$ from the first two terms:

$$\mathcal{O}(PQ \log Q + n^3 PQ + R \log R) = \mathcal{O}(PQ(n^3 + \log Q) + R \log R).$$

If $n^3 \gg \log Q$, then

$$\mathcal{O}(PQ(n^3 + \log Q) + R \log R) = \mathcal{O}(n^3 PQ + R \log R).$$

Otherwise, the safe compact form is

$$\mathcal{O}(PQ(n^3 + \log Q) + R \log R).$$

Table 5 reports the average evaluation times of the explainers. The results indicate that CORTEX has relatively low computational times, even when compared to heuristic explainers, which are generally faster but often less accurate.

Table 5: Comparison of the average evaluation times.

| Explainer | Avg. evaluation time |
|---|---|
| OBS | 4.139 s |
| DDBS | 9.439 s |
| GNN-MOExp | 0.119 s |
| CORTEX | 0.939 s |

## B   Detailed Experimental Setup

CORTEX and SoTA were evaluated on a standard laptop (i.e., Intel Core i7-13th Gen, 10-Core 16-Thread 16GB RAM) as experiments do not require high-performance hardware.

**Spatio-temporal graph construction.** At each time step $t_i$, we define a graph $G_{t_i}$ where each node $u$ represents an EEG channel. We assign to each node a feature vector containing its normalized value at time $t_i$, along with values from the previous $m = 10$ time steps, each lagged by $\ell = 10$. We compute edge weights as the absolute value of the cross-correlation between each pair of EEG channels over a sliding window of the past $s = 5$ seconds. In our experiments, given the signal frequency and sampling rate, the sliding windows used to compute correlations overlap on average by approximately 15% during normal periods and by 90% during seizure events (values can vary depending on the specific time series). To prevent fully-connected graphs, we retain only the top $k_n = 4$ edges per node and symmetrize the adjacency matrix. We select these hyperparameters through patient-specific grid searches and choose the final configuration to maximize oracle accuracy across patients. We encode cross-correlations as edge weights, following prior work on GNNs for EEG data (Tian & Zhang, 2025; Abadal et al., 2025) and to capture temporal variations in inter-channel dependencies. We assign a binary label to each graph based on whether its time window includes any seizure activity. To address class imbalance, we apply non-uniform subsampling: we increase the sampling frequency during seizure periods and adjust it based on seizure duration. We exclude a total of 5% of each seizure window – equally split between the beginning and the end – to avoid unstable transitions that might blur class boundaries. This exclusion improves oracle accuracy while minimizing the amount of discarded data.

**Oracle training.** The design of the oracle does not have a direct influence on the overall pipeline. Its impact on the generated explanations is limited to its final outputs. Therefore, it is only important to ensure its accuracy. Here, we use a simple 2-layer Graph Convolutional Network (GCN) (Kipf & Welling, 2017), where node features of size $d'$ are expanded to $4d'$ through the convolutional layers, and then mapped to the output via a fully-connected layer. A deeper investigation of the architecture would fall outside the scope of this work. Despite being trained on individual temporal snapshots, it is still possible to use our temporal approach for the explainer. To better capture individual variability and improve prediction accuracy, a separate oracle was independently trained for each patient using all EEG recordings for that subject, reserving 10% of the data as a test set for the oracle. The performance metrics across recordings for each patient are reported in Section C.

**Robust candidates identification.** Once the oracle has been trained, we analyze the phase trajectory $(\sigma, v)$ of its sigmoid outputs. This requires setting the parameter $w$, which defines the length of the temporal window used in Equation (4), and the parameter $q$, which determines the fraction of instances considered robust when defining the set $\mathcal{G}'_q$ in Equation (6). These values must be calibrated based on the specific application. To guide this selection, we recommend examining the phase trajectory and the corresponding set of robust candidates $\mathcal{G}'_q$ graphically, as illustrated in Figures 4 and 5. In general, higher values of $q$ result in a larger number of points in $\mathcal{G}'_q$, including points that may be less robust, which can cause the explainer to produce more variable results. In our experiments, we observed that varying $w$ generally yields similar sets $\mathcal{G}'_q$. The main difference lies in the selection of points near the peaks of the sigmoid series: as $w$ increases, the points in $\mathcal{G}'_q$ tend to be located progressively further from the peaks (see Figure 5). Therefore, the choice of $w$ should be guided by the level of robustness required for the specific case study. In our experiments, we set $w = 10$. The set of robust candidates $\mathcal{G}'_q$ is determined for each patient's EEG recording by selecting the $q = 25\%$ of points closest to $(0, 0)$ in terms of Euclidean distance. Although we recommend choosing $w$ and $q$ based on the visual inspection of the phase trajectory in Figures 4 and 5, together with domain knowledge about the required level of temporal stability, it is also possible to examine this choice *a posteriori* through the fidelity values obtained after evaluation. The phase trajectory provides clearer insights into peak localization and series stability, whereas the fidelity analysis can be used as a complementary consistency check. Table 6 reports the fidelity values for the recording `chb03_01` corresponding to the configurations shown in Figures 4 and 5 and additional combinations of the same parameters.

**Multi-objective selection criterion.** Given an anomalous instance $G_i$ from a specific EEG recording, we select the top $k = 5$ counterfactuals by minimizing the objective in Equation (11), which combines dissimi-

larity, temporal distance, and counterfactual robustness. A visual representation of the top-5 counterfactuals is provided in Section E.

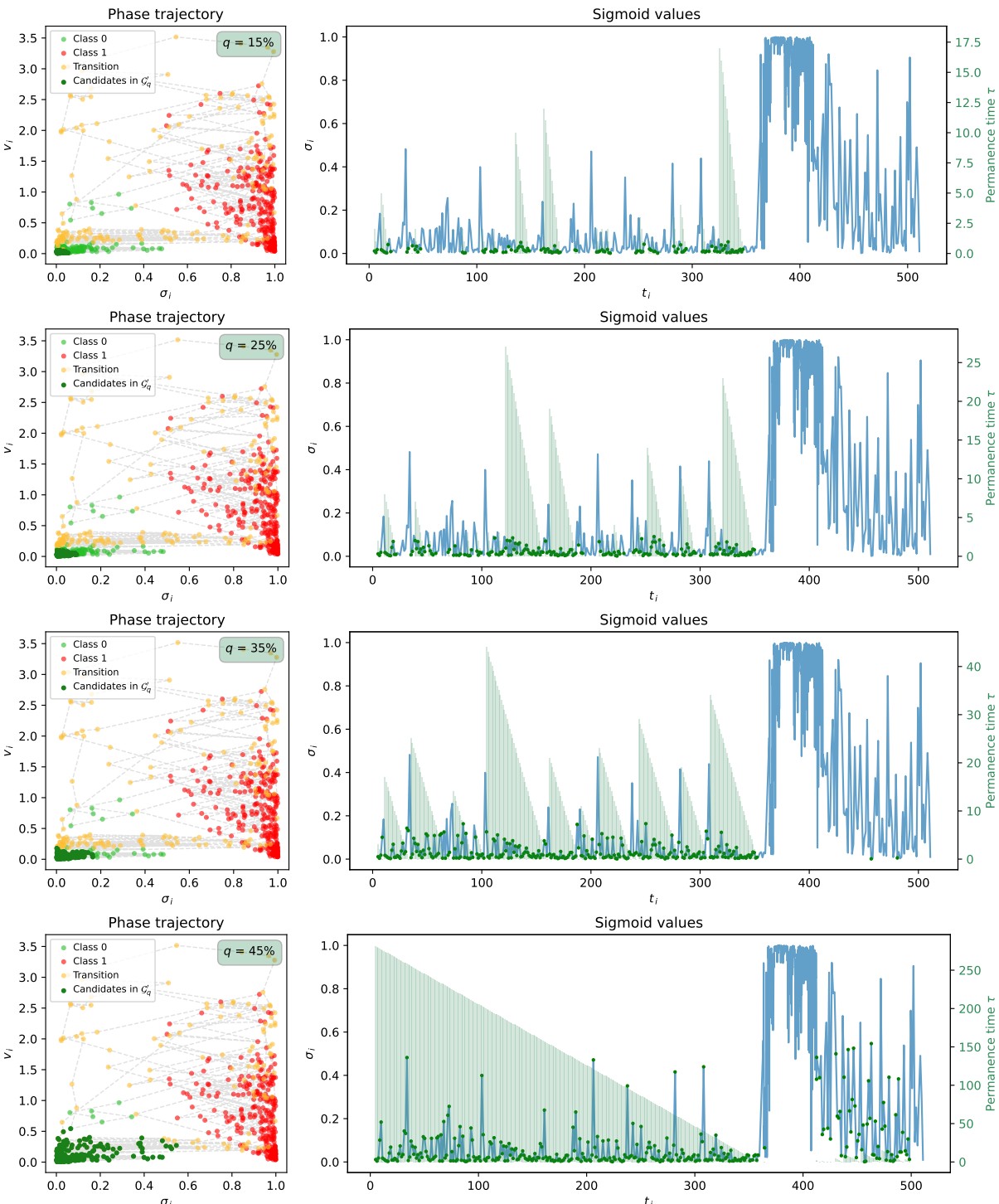

Figure 4: Sigmoid phase trajectory and points in $\mathcal{G}'_q$, as in Figure 3, for $w = 10$ and different values of $q$. Example taken from recording `chb03_01`.

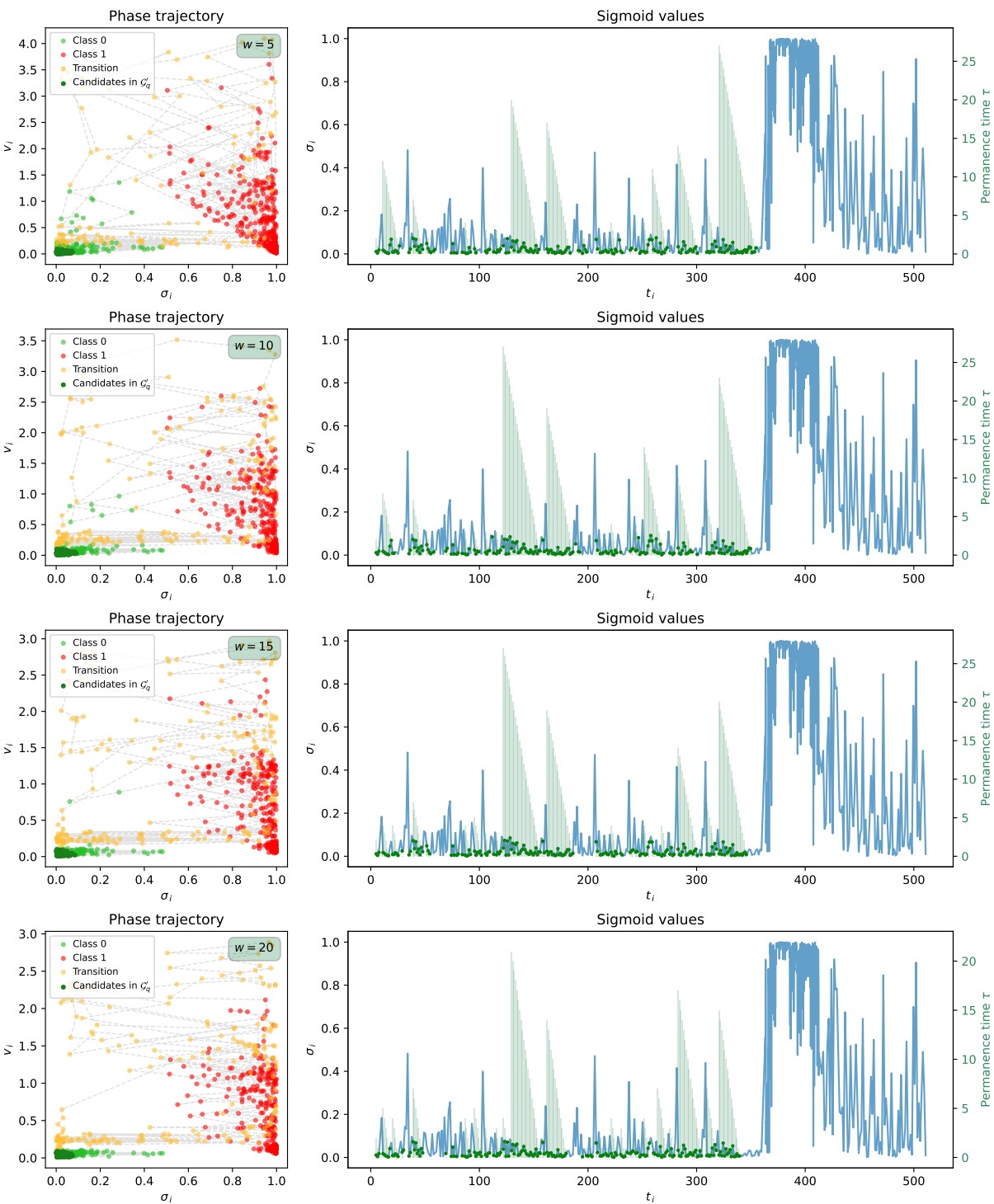

Figure 5: Sigmoid phase trajectory and points in $\mathcal{G}'_q$, as in Figure 3, for $q = 25\%$ and different values of $w$. Example taken from recording `chb03_01`.

Table 6: Fidelity values for different combinations of $w$ and $q$. Example taken from recording `chb03_01`.

|  | $q = 15\%$ | $q = 25\%$ | $q = 35\%$ | $q = 45\%$ |
|---|---|---|---|---|
| $w = 5$ | 0.8484 | 0.8489 | 0.8677 | 0.8584 |
| $w = 10$ | 0.8486 | **0.8953** | 0.8679 | 0.8584 |
| $w = 15$ | 0.8494 | 0.8234 | 0.8677 | 0.8499 |
| $w = 20$ | 0.8595 | 0.8613 | 0.8677 | 0.8492 |

## C   Oracle Evaluation

To assess the performance of the oracle, we use four standard evaluation metrics for binary classifiers: accuracy, F1 score, recall, and precision. Let $TP$ denote the number of true positives, $TN$ the number of true negatives, $FP$ the number of false positives, and $FN$ the number of false negatives, where the positive class corresponds to the anomalous condition. The metrics are defined as follows:

$$\text{Accuracy} = \frac{TP + TN}{TP + TN + FP + FN} \,, \qquad \text{F1 score} = \frac{2 \cdot TP}{2 \cdot TP + FP + FN} \,,$$

$$\text{Precision} = \frac{TP}{TP + FP} \,, \qquad \text{Recall} = \frac{TP}{TP + FN} \,.$$

All these metrics range between 0 and 1, with values closer to 1 indicating better performance.

Figure 6 shows accuracy, F1 score, precision, and recall of the oracles for each patient across individual EEG recordings.

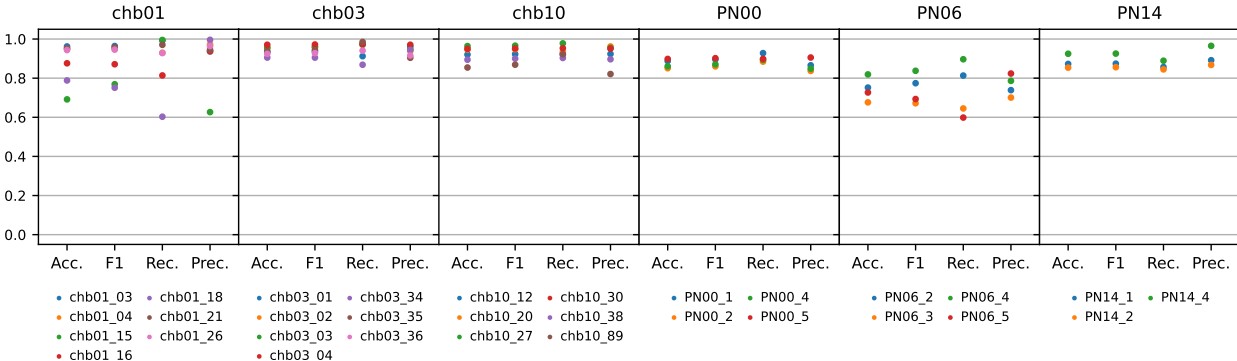

Figure 6: Evaluation metrics of the oracles for each patient (panels) across individual EEG recordings (points).

## D   Explainer Evaluation

The quality of counterfactual explanations can be assessed using different metrics, each capturing a different aspect of the explainer's performance (Prado-Romero et al., 2024b; Guidotti, 2022). Our evaluation criteria include validity, fidelity, dissimilarity, and implausibility, each calculated for every anomalous instance $G \in \mathcal{G}$ and its corresponding counterfactual $G^*$, then averaged over the entire dataset.

**Validity** indicates if the explainer has produced a valid counterfactual, and is defined as

$$\text{Validity}(G, G^*) = \mathbf{1}[\Phi(G) \neq \Phi(G^*)] \,, \tag{12}$$

where $\mathbf{1}[\cdot]$ denotes the indicator function.

**Fidelity** measures how much the explanations are faithful to the oracle, considering its validity. It is defined as

$$\text{Fidelity}_{\text{acc}}(G, G^*) = \mathbf{1}[\Phi(G) = y] - \mathbf{1}[\Phi(G^*) = y] \,, \tag{13}$$

where $y$ is the true label associated to $G$ (Yuan et al., 2023). A value of 1 indicates that both the explainer and the oracle are correct, while a value of 0 or $-1$ indicates an issue with either the explainer or the oracle. By focusing on the predicted probabilities, a more sensitive measure of fidelity can be computed as

$$\text{Fidelity}_{\text{prob}}(G, G^*) = \sigma(G)_y - \sigma(G^*)_y \,. \tag{14}$$

High values indicate better performance, as they reflect a substantial change in the prediction between $G$ and $G^*$. Low values, instead, indicate little or no change, which may suggest that the generated counterfactual is invalid or adversarial. Unless otherwise specified, when we refer to *fidelity*, we mean $\text{Fidelity}_{\text{prob}}$ as defined in Equation (14).

**Dissimilarity** quantifies the closeness between an instance and its counterfactual. Here, $M_\alpha$ serves as the dissimilarity measure, as defined in Equation (7).

**Implausibility** is given by the minimum distance of the counterfactual $G^*$ from the closest example in a reference population $\mathcal{R}$:

$$\text{Implausibility}(G^*) = \min_{G \in \mathcal{R}} D(G^*, G) \,. \tag{15}$$

Here, the distance $D(\cdot, \cdot)$ is again represented by the dissimilarity measure $M_\alpha$. In our setting, when evaluating the implausibility of a counterfactual $G^*_{t_i}$ for an anomalous instance $G_{t_i}$, the reference population $\mathcal{R}$ is chosen to comprise all graphs in the dataset $\mathcal{D}$ observed before $t_i$ that were classified as 0 by the oracle. This ensures that the implausibility reflects how different the counterfactual is with respect to past instances of the target class. The use of the minimum distance, rather than an average, follows the formulation used by Guidotti (2022), where implausibility is interpreted as the deviation from the most similar instance in the reference population.

Ideal values for these metrics are validity and fidelity equal to 1, and implausibility and dissimilarity equal to 0. By design, CORTEX always generates valid and plausible counterfactuals. Therefore, validity and fidelity are only related to the oracle accuracy, while implausibility is always equal to zero. We include these metrics primarily for historical reasons and because they are considered standard in the literature.

In our results, we report validity and fidelity averaged only over correctly classified instances, and dissimilarity and implausibility averaged only over instances where $\text{Fidelity}_{\text{acc}} = 1$.

Table 7 reports the average implausibility and dissimilarity of the top-1 counterfactuals for each individual EEG recording, both computed only for explanations with $\text{Fidelity}_{\text{acc}}$ equal to 1, in order to avoid bias from incorrect counterfactuals. GNN-MOExp is excluded from the table, as its dissimilarity and implausibility are undefined.

Table 7: Average implausibility and dissimilarity of top-1 counterfactuals for individual EEGs. Best values in bold.

| chb01_03 (oracle accuracy: 0.9621) | | | chb01_04 (oracle accuracy: 0.9501) | | | chb01_15 (oracle accuracy: 0.6915) | | |
|---|---|---|---|---|---|---|---|---|
| Explainer | Implausibility | Dissimilarity | Explainer | Implausibility | Dissimilarity | Explainer | Implausibility | Dissimilarity |
| OBS | 1.2003 | 0.4365 | OBS | 1.2775 | 0.3717 | OBS | 1.4889 | 0.4725 |
| DDBS | 0.9309 | **0.2715** | DDBS | 1.1972 | **0.2787** | DDBS | 1.2815 | **0.3156** |
| CORTEX | **0** | 1.7816 | CORTEX | **0** | 1.4246 | CORTEX | **0** | 1.6931 |

| chb01_16 (oracle accuracy: 0.8760) | | | chb01_18 (oracle accuracy: 0.7883) | | | chb01_21 (oracle accuracy: 0.9518) | | |
|---|---|---|---|---|---|---|---|---|
| Explainer | Implausibility | Dissimilarity | Explainer | Implausibility | Dissimilarity | Explainer | Implausibility | Dissimilarity |
| OBS | 1.4973 | 0.4974 | OBS | 0.8399 | 0.1922 | OBS | 0.9665 | 0.5331 |
| DDBS | 1.4046 | **0.4083** | DDBS | 0.8054 | **0.1605** | DDBS | 0.8461 | **0.4181** |
| CORTEX | **0** | 1.6016 | CORTEX | **0** | 0.9082 | CORTEX | **0** | 1.1320 |

| chb01_26 (oracle accuracy: 0.9437) | | |
|---|---|---|
| Explainer | Implausibility | Dissimilarity |
| OBS | 1.1791 | 0.3326 |
| DDBS | 1.0719 | **0.2404** |
| CORTEX | **0** | 1.2709 |

| chb03_01 (oracle accuracy: 0.9336) | | |
|---|---|---|
| Explainer | Implausibility | Dissimilarity |
| OBS | 0.4121 | 0.1101 |
| DDBS | 0.3731 | **0.0773** |
| CORTEX | **0** | 0.4476 |

| chb03_02 (oracle accuracy: 0.9550) | | |
|---|---|---|
| Explainer | Implausibility | Dissimilarity |
| OBS | 0.6800 | 0.2233 |
| DDBS | 0.5273 | **0.1025** |
| CORTEX | **0** | 0.8432 |

| chb03_03 (oracle accuracy: 0.9554) | | |
|---|---|---|
| Explainer | Implausibility | Dissimilarity |
| OBS | 0.8045 | 0.2170 |
| DDBS | 0.7965 | **0.1897** |
| CORTEX | **0** | 0.9130 |

| chb03_04 (oracle accuracy: 0.9708) | | |
|---|---|---|
| Explainer | Implausibility | Dissimilarity |
| OBS | 0.0519 | 0.2992 |
| DDBS | 1.0217 | **0.2422** |
| CORTEX | **0** | 1.1333 |

| chb03_34 (oracle accuracy: 0.9052) | | |
|---|---|---|
| Explainer | Implausibility | Dissimilarity |
| OBS | 0.9624 | 0.2567 |
| DDBS | 0.7707 | **0.1644** |
| CORTEX | **0** | 1.2208 |

| chb03_35 (oracle accuracy: 0.9376) | | |
|---|---|---|
| Explainer | Implausibility | Dissimilarity |
| OBS | 0.8425 | 0.2537 |
| DDBS | 0.6000 | **0.1428** |
| CORTEX | **0** | 1.1962 |

| chb03_36 (oracle accuracy: 0.9246) | | |
|---|---|---|
| Explainer | Implausibility | Dissimilarity |
| OBS | 0.8376 | 0.2489 |
| DDBS | 0.7162 | **0.1630** |
| CORTEX | **0** | 1.2513 |

| chb10_12 (oracle accuracy: 0.9206) | | |
|---|---|---|
| Explainer | Implausibility | Dissimilarity |
| OBS | 0.5573 | 0.1416 |
| DDBS | 0.5312 | **0.1027** |
| CORTEX | **0** | 0.6643 |

| chb10_20 (oracle accuracy: 0.9462) | | |
|---|---|---|
| Explainer | Implausibility | Dissimilarity |
| OBS | 0.5905 | 0.1370 |
| DDBS | 0.4842 | **0.0808** |
| CORTEX | **0** | 0.8033 |

| chb10_27 (oracle accuracy: 0.9641) | | |
|---|---|---|
| Explainer | Implausibility | Dissimilarity |
| OBS | 0.7422 | 0.1894 |
| DDBS | 0.4877 | **0.0813** |
| CORTEX | **0** | 1.1736 |

| chb10_30 (oracle accuracy: 0.9501) | | |
|---|---|---|
| Explainer | Implausibility | Dissimilarity |
| OBS | 0.9513 | 0.2093 |
| DDBS | 0.8693 | **0.1410** |
| CORTEX | **0** | 1.1082 |

| chb10_38 (oracle accuracy: 0.8943) | | |
|---|---|---|
| Explainer | Implausibility | Dissimilarity |
| OBS | 1.0010 | 0.2314 |
| DDBS | 0.8795 | **0.1394** |
| CORTEX | **0** | 1.4398 |

| chb10_89 (oracle accuracy: 0.8544) | | |
|---|---|---|
| Explainer | Implausibility | Dissimilarity |
| OBS | 0.9088 | 0.2243 |
| DDBS | 0.8168 | **0.1633** |
| CORTEX | **0** | 1.1030 |

| PN00_1 (oracle accuracy: 0.8876) | | |
|---|---|---|
| Explainer | Implausibility | Dissimilarity |
| OBS | 1.3047 | 1.0654 |
| DDBS | 1.2586 | **1.0256** |
| CORTEX | **0** | 1.3237 |

| PN00_2 (oracle accuracy: 0.8508) | | |
|---|---|---|
| Explainer | Implausibility | Dissimilarity |
| OBS | 0.9925 | 0.9394 |
| DDBS | 0.8648 | **0.7789** |
| CORTEX | **0** | 0.8519 |

| PN00_4 (oracle accuracy: 0.8608) | | |
|---|---|---|
| Explainer | Implausibility | Dissimilarity |
| OBS | 1.0833 | 0.9085 |
| DDBS | 1.0034 | **0.8186** |
| CORTEX | **0** | 1.1169 |

| PN00_5 (oracle accuracy: 0.8983) | | |
|---|---|---|
| Explainer | Implausibility | Dissimilarity |
| OBS | 0.3960 | 0.2386 |
| DDBS | 0.3816 | **0.2168** |
| CORTEX | **0** | 0.4756 |

| PN06_2 (oracle accuracy: 0.7519) | | |
|---|---|---|
| Explainer | Implausibility | Dissimilarity |
| OBS | 0.0946 | 0.0453 |
| DDBS | 0.0910 | **0.0411** |
| CORTEX | **0** | 0.1386 |

| PN06_3 (oracle accuracy: 0.6764) | | |
|---|---|---|
| Explainer | Implausibility | Dissimilarity |
| OBS | 0.0673 | 0.0438 |
| DDBS | 0.0634 | **0.0367** |
| CORTEX | **0** | 0.2333 |

| PN06_4 (oracle accuracy: 0.8192) | | |
|---|---|---|
| Explainer | Implausibility | Dissimilarity |
| OBS | 0.1979 | 0.0915 |
| DDBS | 0.1917 | **0.0820** |
| CORTEX | **0** | 0.2736 |

| PN06_5 (oracle accuracy: 0.7268) | | |
|---|---|---|
| Explainer | Implausibility | Dissimilarity |
| OBS | 0.1925 | 0.0972 |
| DDBS | 0.1816 | **0.0924** |
| CORTEX | **0** | 0.4209 |

| PN14_1 (oracle accuracy: 0.8727) | | |
|---|---|---|
| Explainer | Implausibility | Dissimilarity |
| OBS | 0.1463 | 0.0470 |
| DDBS | 0.1385 | **0.0404** |
| CORTEX | **0** | 0.2330 |

| PN14_2 (oracle accuracy: 0.8538) | | |
|---|---|---|
| Explainer | Implausibility | Dissimilarity |
| OBS | 0.1368 | 0.1020 |
| DDBS | 0.1350 | **0.0971** |
| CORTEX | **0** | 0.1970 |

| PN14_4 (oracle accuracy: 0.9248) | | |
|---|---|---|
| Explainer | Implausibility | Dissimilarity |
| OBS | 0.6103 | **0.2448** |
| DDBS | 0.6850 | 0.2837 |
| CORTEX | **0** | 1.0545 |

OBS and DDBS consistently show lower dissimilarity scores, but these are often associated with higher implausibility, indicating that the generated counterfactuals are structurally less realistic. CORTEX, instead,

achieves zero implausibility by design, as it searches counterfactuals within the dataset. We stress that implausibility is the primary factor in the evaluation: when implausibility is high, dissimilarity becomes uninformative, as proximity to the original graph cannot compensate for a lack of plausibility.

## E   Visualization of Explanations

Here we provide graphical representations that offer both a global and a detailed view of the topological changes in the counterfactual explanations compared to the original instance. All the visualizations refer to the same anomalous instance from recording `chb03_01`.

Figure 7 provides a summarized view of the top-5 counterfactuals for an anomalous instance by highlighting edges that are consistently added or removed across all of them. This summary helps identify structural changes that appear in most (or all) counterfactuals, which may correspond to meaningful topological changes rather than random variations. Figure 8 details the topology of each of the top-5 counterfactuals, highlighting added, removed, and preserved edges. Finally, Figure 9 offers a more detailed comparison of edge weights between the original graph and its counterfactuals.

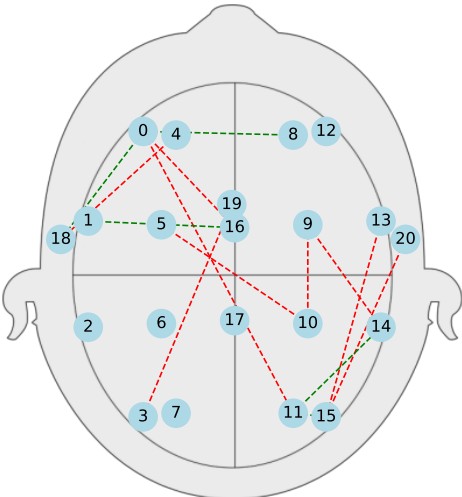

Figure 7: Visualization of edges consistently added (green) or removed (red) across the top-5 counterfactuals for an anomalous instance. Example from recording `chb03_01`.

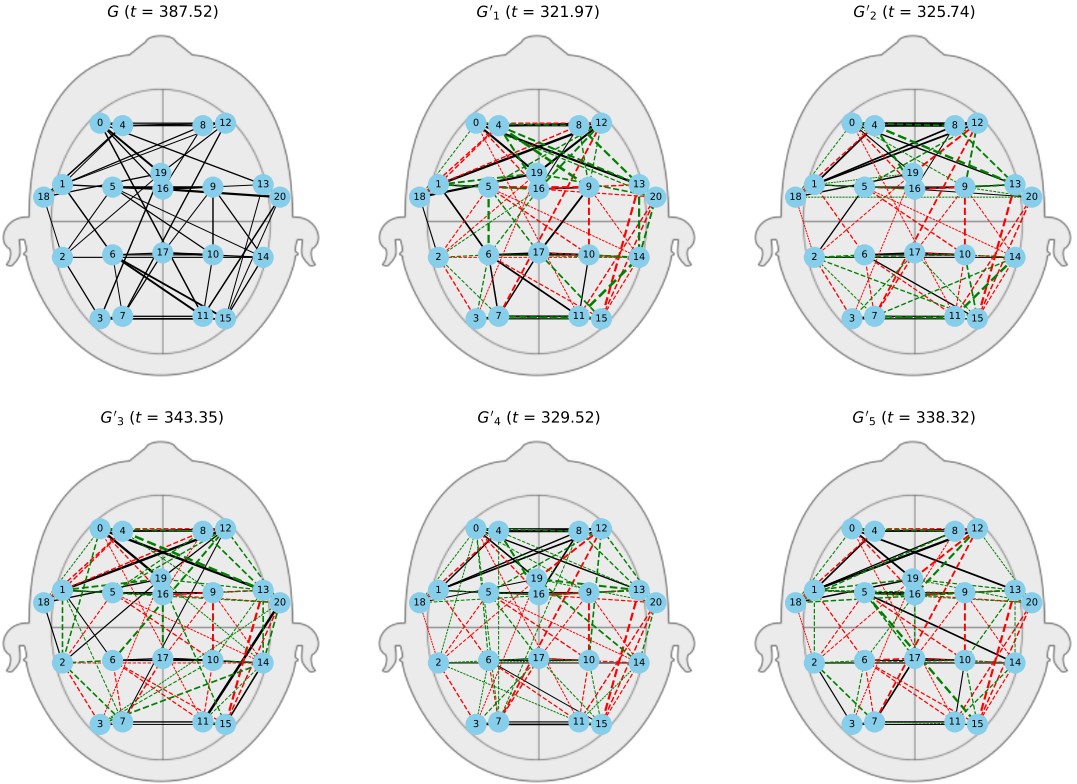

Figure 8: Comparison between an anomalous instance and its top-5 counterfactuals. Black edges are preserved, red dashed edges are removed from the original graph, and green dashed edges are added in the counterfactual. Edge thickness is proportional to the corresponding weight. Example from recording `chb03_01`.

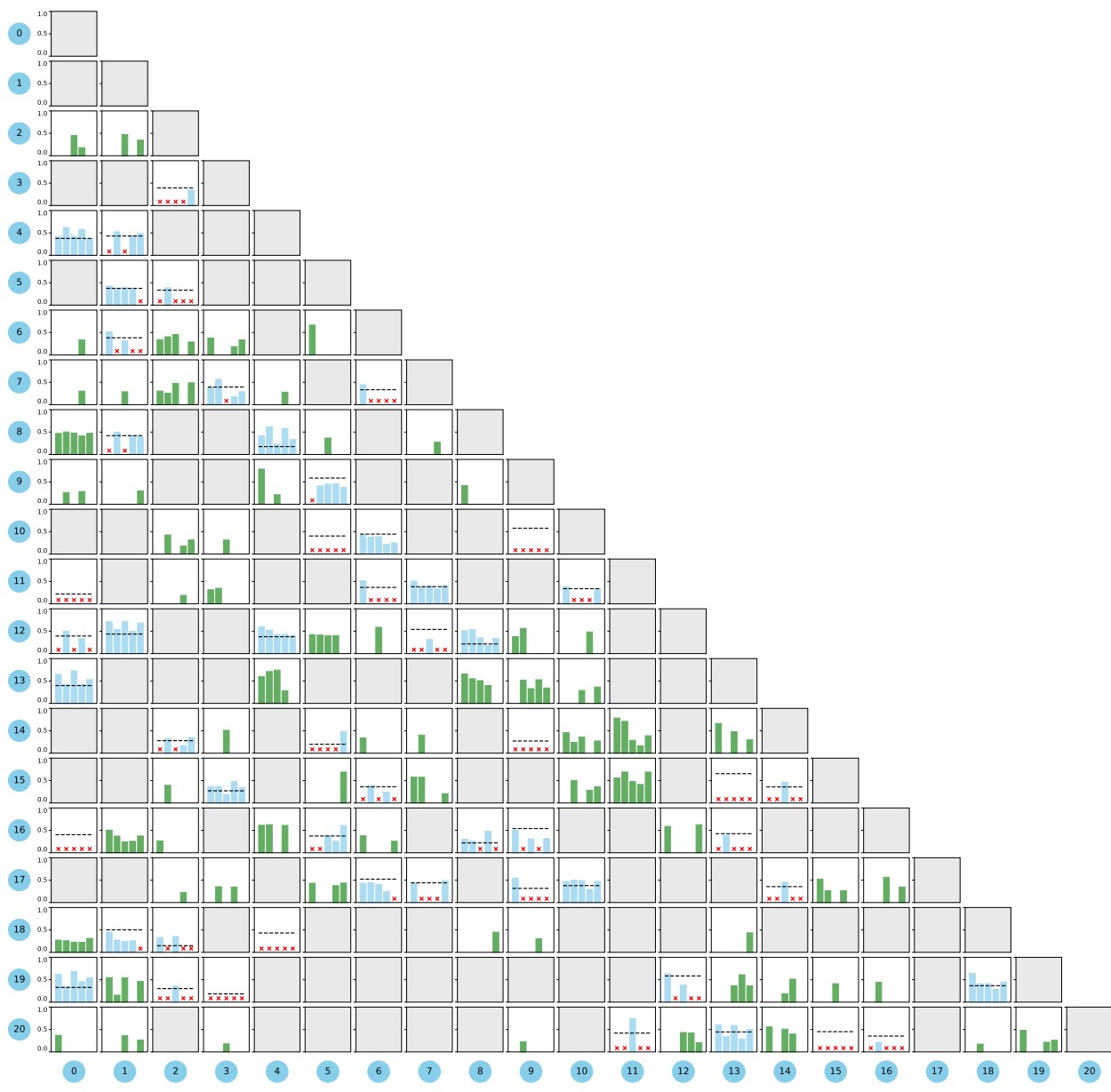

Figure 9: Edge weights of an anomalous instance and its top-5 counterfactuals. Each cell corresponds to an undirected edge between two nodes, with node labels shown along the axes. The dashed line shows the original weight, while vertical bars indicate the corresponding edge weights in the counterfactuals (green bars for added edges, red crosses for removed edges, and blue bars for modified edges). Example from recording chb03_01.

