# OpenReview forum: "COunterfactual Reasoning for Temporal EXplanations: Plausible and Robust Explanations for EEG-Based Seizure Detection"
_TMLR — Accepted by TMLR_

### Review · Reviewer_G9kL · 2025-12-22

**Summary Of Contributions:**

The paper proposed CORTEX, which is a framework for temporal counterfactual reasoning on the Electroencephalography data. The method searching for the relationship in the feature, time, and space on the  EEG spatio-temporal graphs. Compare with the previous feature only work, the method focus on the temporal robustness , that is the counterfacutal state will e stable on along the time axises.

**Audience:**

Yes

**Audience Explanation:**

The findings of this paper would be of interest to the TMLR audience, particularly researchers working on Explainable AI, machine learning for healthcare, and graph neural networks. Most existing counterfactual explanation methods are designed for static data, whereas this paper introduces a framework for dynamic settings. The work makes a methodological contribution by formally defining temporal robustness. In addition, the proposed approach provides an effective way to capture temporal structural dependencies in EEG data.

**Broader Impact Concerns:**

1. **Clinical Over-reliance.**
There is a potential risk that practitioners may interpret the generated counterfactual explanations as direct medical recommendations rather than as decision-support tools. Without careful framing, such explanations could be mistakenly treated as actionable treatment guidance, which may not align with established clinical protocols.

2. **Actionability**
Although the framework retrieves historically observed states, it does not explicitly incorporate actionability constraints. As a result, some suggested counterfactual states—while clinically plausible in retrospect—may not be readily achievable through available medical interventions, external stimulation, or treatment plans in real-world settings.

**Claims And Evidence:**

Yes

**Claims Explanation:**

The paper provide solid mathematircal framework, implement the novel methodology and proved with emplirical results on the clinical dataset.

1. The paper formally defines the key concept of temporal robustness, moving beyond traditional counterfactual methods that focus on static metrics. This claim is supported by the introduction of an instability score ($S_{\Phi}$) and the use of phase trajectory analysis to identify transition regions in model predictions.

2. Empirical Performance. CORTEX is benchmarked against state-of-the-art explainability methods on 31 EEG recordings from the CHB-MIT and Siena Scalp EEG datasets. The results are compelling: CORTEX achieves a 2.73× improvement in validity and a 5.32× improvement in fidelity compared to the second-best methods.


3. Ablation Studies.
The authors provide a detailed ablation analysis of the multi-objective selection criterion (Equation 11). As shown in Table 3, incorporating both the temporal distance term ($M_{\beta}$) and the robustness term ($M_{\gamma}$) leads to significantly higher-quality explanations compared to using structural dissimilarity ($M_{\alpha}$) alone.

4. Interpretability for Clinical Use.
By modeling EEG signals as spatio-temporal graphs, the proposed framework enables visualizations that are directly meaningful for clinical practitioners. The figures clearly illustrate how specific electrode (node) activations and connectivity (edge) changes contribute to transitioning out of a seizure state.

**Requested Changes:**

1. **Generalizability Beyond EEG**.
The authors are encouraged to discuss the generalizability of the CORTEX framework to data types beyond EEG. While the current work is specifically tailored to seizure detection, the underlying principles—modeling multivariate time series as spatio-temporal graphs and searching for temporally robust counterfactuals—may naturally extend to other domains, such as financial time series or industrial sensor monitoring.

2. **Computational Efficiency and Real-World Timing.**
Although a complexity analysis is provided in Appendix A, the cubic dependency on the number of nodes ($n^3$) and linear dependency on dataset size ($P$) may pose challenges for high-density EEG configurations. The authors should report the actual runtime (in seconds or minutes) of the algorithm on the clinical datasets used and include a direct timing comparison with baseline methods (OBS, DDBS, and GNN-MOExp) to better assess real-world feasibility.

3. **Ablation Study on Fidelity.**
While the paper includes phase trajectory plots that qualitatively explain the selection trade-offs for parameters $w$ and $q$, a quantitative ablation study focusing specifically on Fidelity would strengthen the analysis. Similar to the performance comparisons in Table 2 or Figure 6, such a study would clarify how variations in these parameters affect explanatory accuracy.

4. **Clinical Diagnostic Insights.**
Although actionability is excluded from the formal problem definition, the authors argue that search-based methods produce more realistic counterfactuals. The paper would benefit from deeper clinical interpretation of the results, particularly how identified changes—such as specific edge additions or removals—relate to established neurological knowledge. For example, the authors could further elaborate on how the spatio-temporal counterfactual example on Page 2 (involving slow-wave activity in the right temporal lobe) corresponds to the patterns identified by the CORTEX algorithm.

5. **Formatting Improvements.**
To improve clarity and ease of reference, a dedicated figure label or explicit textual reference should be added for the “Static vs. Spatio-temporal” counterfactual example presented on Page 2.

---

> ### Author Response · Authors · 2026-01-31
> **Response to Reviewer G9kL**
>
> Thank you for taking the time to review our paper and for the positive feedback. Below, we respond to the points you raised.
>
> ### Requested Changes:
>
> 1. **Generalizability Beyond EEG**
>    Thank you for highlighting the importance of generalizability to other data types, which is indeed a strength of our approach. We have emphasized this point in the conclusion, noting that although the current study focuses on seizure detection, CORTEX can be applied to other domains with parameters adapted according to domain knowledge and expected outcomes.
>
> 2. **Computational Efficiency and Real-World Timing**
>     We added a table reporting the average evaluation times of all explainers (Table 5):
>     | Explainer | Avg. evaluation time |
>     |-----------|:--------------------:|
>     | OBS       | 4.139 s              |
>     | DDBS      | 9.439 s              |
>     | GNN-MOExp | 0.119 s              |
>     | CORTEX    | 0.939 s              |
>
>     The results indicate that CORTEX maintains relatively low computational times, even compared to less accurate heuristic explainers. This demonstrates that, despite the dependency on the number of nodes (which is relatively low) and on dataset size (which can be controlled), CORTEX remains practical and feasible for real-world evaluations.
>
> 3. **Ablation Study on Fidelity**
>     In the paper, we recommend selecting $w$ and $q$ a priori based on the qualitative analysis presented in Figures 4 and 5 and domain knowledge. However, as you suggested, these parameters can also be selected a posteriori by considering the fidelity values obtained after evaluation. To this end, we added Table 6, which reports the fidelity values for the different combinations of $w$ and $q$:
>     |          | $q=15\%$ | $q=25\%$   | $q=35\%$ | $q=45\%$ |
>     |:--------:|:--------:|:----------:|:--------:|:--------:|
>     | $w=5$    | 0.8484   | 0.8489     | 0.8677   | 0.8584   |
>     | $w=10$   | 0.8486   | **0.8953** | 0.8679   | 0.8584   |
>     | $w=15$   | 0.8494   | 0.8234     | 0.8677   | 0.8499   |
>     | $w=20$   | 0.8595   | 0.8613     | 0.8677   | 0.8492   |
>
> 4. **Clinical Diagnostic Insights**
>    We agree that a deeper clinical interpretation of the results could provide practical guidance. However, such an investigation goes beyond the scope of the current work, which focuses on highlighting structural patterns and differences rather than prescribing interventions.
>
>    The case study we selected is motivated primarily by prior literature on spatio-temporal EEG analysis and brain topological data analysis. Our goal is to uncover meaningful changes, such as specific edge additions or removals, rather than directly relate them to established neurological knowledge. A more thorough clinical analysis would require domain expertise, and the explanations we provide are intended only to support experts with that knowledge. Unfortunately, our team consists of computer scientists with a strong interest in neuroscience, and we deliberately refrain from offering specific medical guidance, as this could be inappropriate or misleading. Our contribution is designed to serve as a supportive tool for clinical experts.
>
> 5. **Formatting Improvements**
>    We added a label and an explicit reference to the "Static vs. spatio-temporal" counterfactual example.
>
> ### Broader Impact Concerns:
>
> 1. **Clinical Over-reliance**
>    We would like to emphasize that our framework does not provide medical recommendations. Its main goal is to reveal structural differences and patterns in the data, which should be carefully analyzed and interpreted by practitioners. We acknowledge the potential risk that counterfactual outputs could be misinterpreted as actionable guidance; to address this, we included a disclaimer in a footnote after Example 1 clarifying that these outputs are intended solely as decision-support tools, not as clinical intervention advice.
>
> 2. **Actionability**
>    Thank you for this observation. Indeed, our framework does not specify how a suggested counterfactual state could be reached through interventions. The goal of our work is to highlight structural patterns and differences in the system rather than to prescribe how to intervene. An interesting avenue for future research is to investigate whether the intrinsic dynamics of the system make some states naturally easier to achieve than others. For example, there may be counterfactual states toward which the system naturally tends to evolve, and others that are intrinsically more difficult to reach. However, fully understanding this relationship is a challenging open problem, which we are still exploring.

---

### Review · Reviewer_tnz1 · 2025-12-28

**Summary Of Contributions:**

This paper introduces CORTEX, a counterfactual explanation framework for EEG-based seizure detection that models multivariate time series as spatiotemporal graphs. CORTEX retrieves stable past graph states as counterfactuals and ranks them using confidence, temporal stability, and structural similarity criteria. Experiments on clinical EEG datasets show that the approach reliably produces valid and plausible counterfactual explanations and reports higher fidelity than existing baselines under the proposed evaluation.

**Audience:**

Yes

**Audience Explanation:**

The paper would be of interest to readers in interpretability and time-series graph modeling, particularly those working on counterfactual explanations and applications in clinical EEG analysis.

**Broader Impact Concerns:**

No. I do not find any broader impact concerns arising from this work.

**Claims And Evidence:**

Yes

**Claims Explanation:**

The claims of the paper are supported by the empirical results and a clearly described methodology. However, the abstract and introduction frame improvements in validity and zero implausibility as evidence of superior modeling over prior work, whereas these properties largely follow from the design choices of CORTEX. I would encourage the authors to reframe the experimental conclusions to better reflect this distinction.

**Requested Changes:**

The paper is generally well written and clearly structured. Despite not working directly in this area, I found the methodology and experimental setup easy to follow. To further improve clarity and strengthen the presentation, I have the following questions and requests for clarification:

**C1.** The ranking in Eqs. 5,6 does not account for the asymmetry or scale differences between $\sigma$ and $v$, where $\sigma$ is a probability-like confidence score and $v$ is a normalized temporal velocity term. Could the authors clarify how sensitive the ranking is to this choice, and what would happen if one term were weighted more heavily than the other?

**C2.** While the preliminaries introduce a general classification setting, the paper only explicitly states that the target task is a binary classification in Section 4. It would improve clarity to state this explicitly in the preliminaries so as not to confuse readers.

**C3.** The motivation for using graph-based counterfactuals and spatiotemporal graph representations is not fully discussed. A brief justification of this choice, along with a discussion of alternative representations or counterfactual formulations and why they may be less suitable, would strengthen the motivation.

**C4.** It is unclear whether the proposed counterfactual explanations are intended for online or real-time use. Since temporal robustness relies on multiple predictions over time, clarifying whether this framework is purely offline or whether it is meant to support real-time seizure detection scenarios would be helpful, particularly given the potential real-world implications of the latter.

**C5.** Please refer to the explanation provided for “Are the claims made in the submission…” and consider updating the abstract and introduction to better align the framing of the experimental conclusions with the design choices of the proposed method.

---

> ### Author Response · Authors · 2026-01-31
> **Response to Reviewer tnz1**
>
> Thank you for the time spent reviewing our paper and for the positive feedback. Below, we provide our responses to the comments.
>
> **C1.** We agree that, in principle, different relative weightings between $v$ and $\sigma$ could be considered. We did not introduce any weighting in our implementation, as there is no principled way to choose one a priori, and we observed that this choice has minimal impact on the results. Indeed, the distance $\delta$ is used to select points closest to the origin, where both $v$ and $\sigma$ are small and candidate points are densely distributed. For this reason, in this region different weightings produce similar rankings, as the relative ordering of points is only weakly affected. Differences due to weighting become more noticeable only for points farther from the origin, which are excluded from the selection and therefore do not influence the results.
>
> **C2.** We clarified that the target task is binary classification right from the abstract and in the preliminaries under Example 1. Additionally, we mentioned the possible extension to multi-class settings in the conclusions.
>
> **C3.** We have revised the introduction to clarify the motivation for using spatio-temporal graph representations and graph-based counterfactuals. This choice allows us to explicitly encode semantic relationships among variables and their temporal interactions, which are otherwise only implicitly captured in standard tabular or sequence-based multivariate time-series representations. By doing so, we enable the definition of structured counterfactuals that account for meaningful dependencies across features, rather than reflecting a preference for a specific modeling paradigm. If you have specific alternative formulations or methods in mind, we would be happy to further discuss them.
>
> **C4.** In our experiments, anomalies have already been labeled, so CORTEX operates offline, analyzing EEG recordings a posteriori and considering the full temporal evolution of each anomaly to highlight structural differences. In our view, CORTEX is a tool that can assist clinicians in a "debugging" process, helping them understand how to transition from an epileptic state to a normal state and potentially prevent recurrence.
> While real-time anomaly detection is outside the scope of this work, the CORTEX methodology itself is almost agnostic to offline or online operation: as long as anomalous segments are flagged by a detector and past stable conditions are available, it can generate explanations and guide recovery toward normal states, regardless of whether the data are acquired offline or in streaming mode. The only modification needed is related to the definition of $v_i$ in Equation 4, which currently depends on future time steps. In a streaming context, $v_i$ could instead be computed using only past time steps, or replaced by an alternative measure of temporal instability, potentially based on predictions of future behavior. We have mentioned this point in Section 5.
>
> **C5.** We agree that the improvements in validity and zero implausibility result from CORTEX's design choices. We have revised the abstract and introduction to clarify this point.
> Regarding the comparison with other benchmarks, we want to point out that counterfactual explanation methods should not be treated as a one-size-fits-all paradigm. While this paradigm might be interesting for non-critical scenarios, some hard constraints remain essential in safety-critical contexts. Unfortunately, we haven't come across other counterfactual methods in our same scenario, and we had to evaluate against other "more generic" search-based or heuristic-based approaches. This leaves us excited to contribute to the literature with the first method.

---

### Review · Reviewer_ggeM · 2026-01-23

**Summary Of Contributions:**

Summary:
Authors propose a temporally aware counterfactual explanation framework for EEG-based seizure detection. The method is post-hoc (given a trained classifier) and relies on constructing graphs from temporal windows of EEG signals. The key idea is to explicitly incorporate temporal robustness into the counterfactual search by identifying candidate counterfactuals based on the oscillation/uncertainty of classification outputs over time, and selecting counterfactual graphs among the most temporally stable ones. Experiments are conducted on two public EEG datasets comprising 36 patients in total. The proposed approach consistently outperforms baseline counterfactual methods.

Strengths:
- The paper is generally well written, with clear organization and a reasonably clear methodology. Reproducibility is acceptable (code availability would be a plus)
- The motivation of counterfactual explanations for potential intervention is relevant and interesting (although this aspect is introduced relatively late in the manuscript)

Weaknesses:
- While the work is framed as counterfactual reasoning for temporal graphs, the proposed methodology is strongly tailored to EEG signals and seizure detection. It is a significant limitation that should be clearly acknowledged and motivated earlier (see major comments).
- The framework is evaluated on a single application with two datasets (31 subjects independently assessed). Seizure detection can be addressed using other physiological signals and settings; it would be useful to discuss whether the proposed temporal robustness formulation could extend to other applications.
- The experimental setting is unclear in several places: the paper mentions 36 subjects initially, but tables report results for 31; metrics are not always clearly defined; and it is challenging to assess comparisons with baselines fairly. The authors argue that learning-based baselines are not applicable because each patient is treated as a separate dataset, but it is unclear why a leave-one-subject-out (train on n-1, test on 1) setup would not be feasible (is inter-subject variability and domain gap in EEG too substantial?).
- Two hyperparameters are set by “examining the data” (Appendix B), which is a strong limitation, especially since the authors state these parameters are application-specific. The method's sensitivity to these choices is not sufficiently analyzed.

**Additional Comments:**

Major comments:
- The motivation in the introduction is rather high-level and not fully convincing. To me, the crucial motivation is actually discussed in the first paragraph of Section 4 (Context) and it should be moved to the introduction.
- In Definition 2 and the definition of a time graph, the time graph is defined only at a single time t_i, whereas Figure 1 suggests that a time graph corresponds to a set of graphs G_t_j over all t_j < t_i. Similarly, the instability score is defined as a function of a single time index. This inconsistency makes the notion of temporal robustness difficult to fully grasp.
- In Equation (4), the computation of v_i uses future time points relative to i. While this is acceptable in a post-hoc setting, as authors state “Note that this computation does not violate causality, because all values used to compute vi have already been observed in the time series.” But it does violate temporal causality.
- Since this is a post-hoc method, it would be useful to discuss whether a real-time variant could be designed, e.g., using only past information to compute v_i?
- Clarify whether the temporal windows used around t_i overlap with previous windows. While values are provided in Appendix B, the effective temporal span depends on the sampling rate and should be explicitly discussed.
- In Figure 3, red points appear to be more separated along the sigma_i axis, suggesting that sigma_i may correlate more strongly with the class than v_i. Given this, is Euclidean distance the most appropriate choice? A weighted or anisotropic distance (e.g., ellipsoidal) might be more suitable?
- Although the details of parameters w (Equation 4) and q (Equation 6) are provided in Appendix B, a brief summary of how they were chosen should be included in the main text.
- Section 6.2 is difficult to follow. The authors mention two datasets with 22 and 14 patients, then select three patients from each, then state that each patient is treated as a separate dataset, and finally report results on 31 recordings. This pipeline needs to be clarified.
- If validity is defined as the counterfactual belonging to a different class than the original sample, and fidelity is reported separately, statements such as “2.73× improvement in validity and 5.32× in fidelity” are confusing. Percentage-point improvements would be more interpretable. Moreover, reporting validity values of 1.0 everywhere raises questions about the fairness of the comparison. Aggregated statistics across subjects would be more informative (along with statistical significance testing, or at least reporting the stds).

Minor comments:
- In the illustrative scenario in the introduction, the phrase “average slow-wave activity” implicitly assumes a temporal window. While I agree that the proposed method handles temporal information more explicitly, this example might be misleading.
- In Figure 3: add a color legend directly in the figure, not only in the caption.

**Audience:**

Yes

**Audience Explanation:**

- The paper introduces an interesting and timely direction by extending counterfactual explanations to temporal settings (likely to be of interest to the audience working on interpretability, time-series modeling, and graph-based learning)
- The methodological idea of incorporating temporal robustness into counterfactual search is relevant beyond the specific application. But the current contribution is somewhat limited by its strong tailoring to a single application (EEG-based seizure detection). Demonstrating the framework's relevance to additional temporal or spatio-temporal domains would significantly broaden its appeal and impact.

**Broader Impact Concerns:**

No concerns (current broader impact is helping to understand the mechanism and/or the design of intervention treatment for EEG-based seizure detection).

**Claims And Evidence:**

No

**Claims Explanation:**

- While the proposed idea is interesting and the methodology is generally well motivated, the empirical evidence does not fully support the claims made in the paper.
- Important methodological choices are insufficiently justified.
- The experimental setup lacks clarity and consistency (dataset splits, subject counts, metric definitions), which makes it difficult to confidently interpret the reported improvements.

**Requested Changes:**

Critical requested changes:
- Clarity of the experimental setting
- 1 more suitable application from tomporal series graphs (maybe at least another signal-based seizure detection, ECG?)

To strengthen the work:
- Rewrite (mostly reorganize) context / motivation / related work
- More discussion / limitations (sensitivity to the specific application, sensitivity to w and q parameters, sensitivity to sampling rate and other preprocessing steps of data)/

(see weaknesses and major comments for more details)

---

> ### Author Response · Authors · 2026-01-31
> **Response to Reviewer ggeM - Part 1**
>
> Thank you for the effort made to review our paper. We provide our point-by-point responses below.
>
> ### Strengths
> 1. Thank you for the positive feedback. We would like to clarify that the code was already made available in the initial submission through an anonymous GitHub link.
>
> 2. As suggested in *Major Comment 1*, we have clarified in the introduction that counterfactual explanations can guide potential interventions by identifying minimal changes that transition a system from an anomalous to a normal state, here illustrated with epileptic seizure activity.
>
>
> ### Weaknesses
> 1. The focus on EEG-based seizure detection is explicitly stated in the title, abstract, and introduction. A discussion of the rationale behind this choice is provided in our response to *Requested change 2*.
>
> 2. EEG signals are among the most commonly used data types for epileptic seizure detection, but other physiological signals (such as heart rate, blood pressure, blood oxygenation, eye movements, and others) can also be used. Both EEG and heart rate data are high-frequency time series that can capture rapid physiological changes. Converting a time series into a graph is particularly meaningful for EEG signals, as they reflect the brain's inherent topology. We argue that EEG signals are more directly and specifically related to seizure activity, whereas other physiological signals, while potentially informative for seizure detection, may also reflect a variety of unrelated physiological processes.
>
>     The reasoning behind our approach, and in particular the notion of temporal robustness, is generally applicable to any type of time series data. The main differences could lie in practical implementations. As clarified in the conclusions, our approach could in principle be extended to other physiological signals as well as to time series applications in other domains.
>
> 3. It is correct that the two datasets include a total of 36 subjects. However, since patients differ in their EEG characteristics, we chose to treat each patient independently and design patient-specific oracles to maximize the accuracy and reliability of patient-level explanations. To ensure sufficient data for each patient, we select three patients from each dataset (six in total) with the highest number of recordings and seizures, and consistent EEG channels. This results in 31 EEG recordings, providing a dataset that remains large and high-quality, suitable for robust patient-specific analyses. We have clarified this point in Section 6.1.
>
>     Given this intersubject variability, as you pointed out, using learning-based baselines with a leave-one-subject-out approach would not be the best solution. In that scenario, the explainer would primarily learn to provide explanations based on average, general patterns across patients and recordings. In practice, these explanations would be less reliable and less useful, as they reflect average patterns across patients rather than the patient's own EEG patterns, which should form the basis of the explanation. We included this observation in Section 6.2.
>
>     As for the metrics, the explainer evaluation metrics were defined in the Appendix of the original submission. For completeness, we have added the definition of the oracle evaluation metrics. We are available to provide additional details on the explainer evaluation metrics, if necessary.
>
> 4. In the revised version, we have added Table 6, which reports fidelity values for different combinations of $w$ and $q$. This new table provides a more detailed view of parameter sensitivity.
>
>     |          | $q=15\%$ | $q=25\%$   | $q=35\%$ | $q=45\%$ |
>     |:--------:|:--------:|:----------:|:--------:|:--------:|
>     | $w=5$    | 0.8484   | 0.8489     | 0.8677   | 0.8584   |
>     | $w=10$   | 0.8486   | **0.8953** | 0.8679   | 0.8584   |
>     | $w=15$   | 0.8494   | 0.8234     | 0.8677   | 0.8499   |
>     | $w=20$   | 0.8595   | 0.8613     | 0.8677   | 0.8492   |

---

> > ### Author Response · Authors · 2026-01-31
> > **Response to Reviewer ggeM - Part 2**
> >
> > ### Requested Changes
> >
> > **Critical requested changes**
> > 1. We clarified the distinction between "patient" and "recording", detailed the selection procedure for the 31 recordings in Section 6.1, and discussed the choice of hyperparameters. As for the metrics, we added the definition of the oracle evaluation metrics.
> >
> > 2. The main reason for focusing on EEG data is that, during the design of our explainer, we found that suitable datasets for graph-based anomaly detection in multivariate time series remain limited [1]. A suitable dataset for our framework must meet the following criteria. First, the series should be homogeneous and lend itself to a graph representation. Second, the number of features (nodes) should remain manageable to keep explanations interpretable (we argue that the ideal range is around 15-30 features, so that the graph counterfactual is readable). Finally, we prefer datasets with sequential anomalies, as these are the focus of our analysis.
> >
> >     As a first step, we searched for time series in the GNN literature, as they are generally suitable for graph-based analysis and are natural candidates for our work. We examined the datasets collected in [1], focusing specifically on anomaly detection. With few exceptions of papers that do not provide datasets, two main groups emerge. One subset focuses on EEG signals, and a larger subset analyzes industrial datasets including SWaT (51 features), WADI (93 features), SMAP (55 features), and MSL (36 features). All these datasets have sequential anomalies. However, most of these non-EEG series have an excessively large number of features, which leads to very large graphs. As a consequence, the resulting graph counterfactuals become too difficult to interpret, ultimately limiting their applicability.
> >     Even accepting their complexity, these datasets have been subject to recent criticism in the literature. For example, [2] points out that they are flawed: in some cases, the anomalies do not correspond to pattern changes in the series or shifts in correlations, but are simply flat or mislabeled signals. For these reasons, we excluded them from our study.
> >
> >     As a second step, we expanded our search to datasets beyond those used for graph-based analysis, and we examined the papers in [3]. Many of these datasets are either univariate, have too few or too many features for meaningful graph counterfactuals (an ideal range would be 15-30 features, as noted above), or consist of heterogeneous series (rarely studied with GNN models). As a result, they are not suitable for our framework.
> >
> >     Given these considerations, EEG data for seizure detection was the most appropriate choice for our study. This is also supported by a line of research showing that EEG signals are analyzed not only with GNNs, but also with other graph-based and topological techniques, indicating that encoding them as graphs is meaningful (see [4]-[6] for example). Considering the limitations of the other datasets and the post-hoc nature of our analysis, a full exploration of new datasets or training models from scratch is beyond the scope of this work. We plan to do so in the future.
> >
> >     [1] Corradini et al., "A systematic literature review of spatio-temporal graph neural network models for time series forecasting and classification", Neural Networks (2026)
> >     [2] Wu and Keogh, "Current Time Series Anomaly Detection Benchmarks are Flawed and are Creating the Illusion of Progress", IEEE Transactions on Knowledge and Data Engineering (2023)
> >     [3] Jia et al., "Deep anomaly detection for time series: A survey", Computer Science Review (2025)
> >     [4] Ling et al., "Topological data analysis in EEG signal processing: a review", Communications in Mathematical Biology and Neuroscience (2025)
> >     [5] Tian et al., "EEG-based epilepsy detection with graph correlation analysis", Frontiers in Medicine (2025)
> >     [6] Abadal et al., "Graph neural networks for electroencephalogram analysis: Alzheimer's disease and epilepsy use cases", Neural Networks (2025)
> >
> > **To strengthen the work**
> > 1. We clarified context and motivation in the introduction. Regarding the related work, could you please indicate if there are particular aspects you would like us to modify or expand?
> >
> > 2. We discussed the sensitivity to the parameters $w$ and $q$ in the Appendix. In the conclusions, we clarify that this calibration is currently tailored to our datasets and specific case study, and exploring more general or automated strategies will be part of future work.
> >
> >     For the other preprocessing steps (described in "Spatio-temporal graph construction" and "Oracle training" in Appendix B), parameters were selected to maximize oracle accuracy. Changes in the sampling rate affect only the number of examples per class; our only adaptation was to ensure a balanced dataset, thereby avoiding the need for more complex oracle-level techniques beyond the scope of our work.

---

> > > ### Author Response · Authors · 2026-01-31
> > > **Response to Reviewer ggeM - Part 3**
> > >
> > > ### Additional comments
> > >
> > > **Major comments**
> > > 1. We added a clarification in the introduction stating that our work aims to identify the changes required to transition from an anomalous to a normal state, ending the epileptic episode and restoring normal activity. For readability, we did not move the first paragraph of Section 4 to the introduction, but we hope this clarifies the motivation.
> > >
> > > 2. Thank you for pointing out the inconsistency with the notation in Figure 1. Each time graph $G_{t_i}$ is defined at a single time index $t_i$. We revised Figure 1 to avoid confusion.
> > >
> > >     As for the instability score $M_\gamma$, it is evaluated for each graph $G_{t_j} \in \mathcal{G}^\prime_q$. The scores $M_\gamma(G_{t_j})$ depend on the entire set $\mathcal{G}^\prime_q$ and on the hyperparameters $w$ and $q$. We clarified this point in the text under Eq. 10.
> > >
> > > 3. The computation of $v_i$ in Eq. 4 relies on future time points relative to $i$. This choice is consistent with our post-hoc analysis, where the full series is available and instability can be quantified based on the observed evolution of the signal. We acknowledge that this would not be applicable in a streaming setting, as it would require access to future observations not available at inference time. In that case, $v_i$ can be defined using past time steps, or replaced by a proxy for temporal instability based on predictions of future behavior, an interesting direction for future work.
> > >
> > > 4. Although implemented offline, CORTEX could be adapted for real-time settings. This would require an oracle that detects anomalies in real time and access to previous normal data. The main modification concerns the definition of $v_i$ (as noted above, $v_i$ could be computed using only past time steps, or replaced by an alternative measure of instability based on predictions). We added this explanation in Section 5.
> > >
> > > 5. We added details on the overlap of windows used to compute correlations for edge weights in the paragraph "Spatio-temporal graph construction" in Appendix B. The overlap arises because each window needs a sufficient number of points for meaningful correlations, and some overlap is needed to avoid discarding too much data. In our experiments, the windows overlap on average by approximately 15% during normal periods and by 90% during seizures, with exact values varying slightly across series.
> > >
> > > 6. Due to their definitions, $\sigma$ primarily captures differences between classes (points of different classes tend to be separated along the $\sigma$ axis), whereas $v$ distinguishes between transitions across classes and within the same class. This explains why the distribution of points varies across different regions of the plot.
> > >
> > >     We used an unweighted Euclidean distance, as there is no principled way to select a weighting for $v$ and $\sigma$ a priori. We noticed that this choice has minimal impact because the distance $\delta$ is used to identify points closest to the origin, where points are densely clustered. In this region, the relative ordering of points is largely insensitive to the weighting: differences become noticeable only for points farther from the origin, which are excluded from selection and so do not affect the final results.
> > >
> > > 7. We added a brief summary in the "Robust Counterfactual Identification" paragraph in Section 5, with a reference to the Appendix for further details.
> > >
> > > 8. We believe that the confusion arises from the distinction between "patient" and "recording". As discussed in *Weakness 3*, to ensure a valid patient-level analysis, we selected a subset of 6 patients, prioritizing those with the highest number of recordings and seizures, and with consistent EEG channels. This results in 31 EEG recordings, whose IDs are listed in Tables 1 and 2. We clarified this point in Section 6.1. All results are reported for these 31 series, each analyzed using patient-specific oracles.
> > >
> > > 9. We modified statements such as "2.73$\times$ improvement in validity and 5.32$\times$ in fidelity" to "3.73$\times$ higher validity and 5.32$\times$ higher fidelity". We prefer not to use percentage-point improvements, as "+273% improvement in validity and +532% in fidelity" could be misleading. We hope that this is clearer.
> > >
> > >     The fact that validity is always equal to 1 is a result of CORTEX's design, as its search is limited to valid counterfactuals. This behavior is inherent to the method and does not relate to fairness of comparison. We clarified this point in the abstract and introduction to avoid confusion.
> > >
> > >     In the revised version, we report aggregated statistics across subjects for each method, including mean and standard deviation of validity and fidelity, along with the paired two-sample $t$-test comparing each baseline with CORTEX (Table 3).
> > >
> > > **Minor comments**
> > > 1. We do not understand what your concern is. Could you please clarify?
> > >
> > > 2. We added a color legend in Figures 3, 4 and 5.

---

### Author Response · Authors · 2026-01-31
**General response**

We thank the reviewers for their feedback. We have uploaded a revised version of the manuscript with changes highlighted.

---

### Decision · Action_Editor_PqqT · 2026-02-28

**Recommendation:** Accept with minor revision

**Additional Comments:**

I invite the authors to address the following points emerged by the reviewers:
1) include the clarification regarding the offline vs. streaming setting in the final version.
2) discuss the following statement "Without broader validation or clearer evidence of cross-domain robustness, it is difficult to assess the method’s impact beyond this specific setting."
3) provide some additional evidence that the approach sensiby extends to other domains.

**Audience:**

Yes

**Audience Explanation:**

Two rewievers agree that the paper presents an interesting methodological contribution, which remains strongly tailored to a specific application (EEG-based seizure detection).

**Claims And Evidence:**

Yes

**Claims Explanation:**

According to the reviewers, the new submitted version clarifies several aspects of the experimental setup and the added ablation studies helped to clarify the behavior and contribution of the proposed temporal robustness component. The same applies to computational efficiency and parameter sensitivity because the authors provided the requested runtime comparisons.

---

> ### Author Response · Authors · 2026-03-07
> **Response to AE**
>
> Dear AE,
> Thank you for accepting our paper with minor revision and for your suggestions.
> We have revised our manuscript and uploaded the camera-ready version.
> Kind regards.

---

> > ### Comment · Action_Editor_PqqT · 2026-03-07
> > **Thank you so much**
> >
> > I'll make the last check by today
> > All the best